

# BuRNN (v1.0): A Data-Driven Fire Model

Seppe Lampe[1], Lukas Gudmundsson[2], Basil Kraft[2], Stijn Hantson[3], Douglas Kelley[4],
Vincent Humphrey[5], Bertrand Le Saux[6], Emilio Chuvieco[7], and Wim Thiery[1]

[1]Department of Water and Climate, Vrije Universiteit Brussel, Brussels, Belgium
[2]Institute for Atmospheric and Climate Science, ETH Zürich, Zürich, Switzerland
[3]School of Sciences and Engineering, Universidad del Rosario, Bogotá, Colombia
[4]Centre for Ecology and Hydrology, Wallingford, UK
[5]Federal Office of Meteorology and Climatology MeteoSwiss, Zürich, Switzerland
[6]Φ-lab, European Space Agency, Frascati, Italy
[7]Environmental Remote Sensing Research Group, Universidad de Alcalá, Alcalá de Henares, Spain

**Correspondence:** Seppe Lampe (seppe.lampe@vub.be)

**Abstract.** Fires play an important role in the Earth system but remain complex phenomena that are challenging to model numerically. Here, we present the first version of BuRNN, a data-driven model simulating burned area on a global $0.5° \times 0.5°$ grid with a monthly time resolution. We trained Long Short-Term Memory networks to predict satellite-based burned area (GFED5) from a range of climatic, vegetation and socio-economic parameters. We employed a region-based cross-validation strategy
to account for the high spatial autocorrelation in our data. BuRNN outperforms the process-based fire models participating in ISIMIP3a on a global scale across a wide range of metrics. Regionally, BuRNN outperforms almost all models across a set of benchmarking metrics in all regions. However, in the African savannah regions and Australia burned area is underestimated, leading to a global underestimation of total area burned. Through eXplainable AI (XAI) we unravel the difference in regional drivers of burned area in our models, showing that the presence/absence of bare ground and C4 grasses along with the fire
weather index have the largest effects on our predictions of burned area. Lastly, we used BuRNN to reconstruct global burned area for 1901-2019 and compare the simulations against independent long-term historical fire observation databases in five countries and the EU. Our approach highlights the potential of machine learning to improve burned area simulations and our understanding of past fire behaviour.

## 1 Introduction

Fire plays an important role in the Earth system by influencing ecosystem dynamics, biogeochemical cycles and atmospheric composition (Bowman et al., 2020). Fires drive ecosystem dynamics by affecting plant evolution (Simon et al., 2009), vegetation species composition and the physical, chemical and biological properties of soils (McLauchlan et al., 2020). Many of these ecosystem characteristics in turn also shape fire behaviour (Archibald et al., 2018). Emissions from vegetation fires affect the radiative balance of the Earth as the gases ($H_2O$, $CO_2$) trap energy through the greenhouse effect, while the aerosols reduce the
amount of solar radiation that reaches Earth's surface (Bowman et al., 2009; Ward et al., 2012). Smoke of fires affects a wide range of systems including the radiative balance (Hodzic et al., 2007; Chakrabarty et al., 2023), plant fertilization (Fritze et al.,



1994; Bauters et al., 2021), albedo (Beck et al., 2011; Veraverbeke et al., 2012) and air quality (Carvalho et al., 2011; Chen et al., 2017). Fires act as a big natural hazard and can also precondition post-fire hazards such as floods, landslides and large-scale erosion (Zscheischler et al., 2020; Jacobs et al., 2016; Girona-García et al., 2021; Brogan et al., 2017; Shakesby, 2011).

Global observations of fire activity are typically provided by satellite products. However, these observations contain substantial uncertainties due to their spatial resolution, cloud cover and temporal resolution affecting their ability to detect small and short-lived fires. Moreover, smoke, rapid regrowth and obscuration by unburned vegetation further complicates satellite-based fire detection. Nonetheless, satellites provide the most reliable estimates of global fire activity to date. Vegetation fires burn approximately 3.5-4.5 million km$^2$ of surface area per year (Giglio et al., 2018; Lizundia-Loiola et al., 2020) and emit between

1.8 and 3.0 Pg Cyr$^{-1}$ (Lizundia-Loiola et al., 2020; van der Werf et al., 2017). More recent estimates from Global Fire Emissions Database version 5 (GFED5) however suggest the amount of surface area burned per year to be around 6.5-9.5 million km$^2$ (Chen et al., 2023b) with an emission of 2.9-3.7 Pg Cyr$^{-1}$ (Chen et al., 2023b), comparable to around 20-30% of the annual emissions from anthropogenic greenhouse gases (Friedlingstein et al., 2025). Fires thus play an active role in our Earth system. Yet, despite their key role, it is not fully understood and quantified how socio-economical development and climate change have affected fire occurrence in the past, and how these will affect future fire dynamics.

have affected fire occurrence in the past, and how these will affect future fire dynamics.

To understand how climate change and socio-econmic conditions affect vegetation fires, researchers typically model fire activity with fire-coupled Dynamic Global Vegetation Models (DGVMs) (e.g., Burton et al., 2024; Park et al., 2024). These process-based fire models simulate vegetation fires as a function of vegetation characteristics, weather, socio-economic conditions, lightning and land use (Hantson et al., 2016). Vegetation dynamics are typically supplied by the DGVM, while the other

factors are provided as inputs derived from climate and integrated assessment models (Frieler et al., 2024). From these drivers, most fire models simulate ignitions (natural + anthropogenic), fuel (dry vegetation), fire spread and fire suppression, which are then transformed to fire characteristics such as burned area, fire intensity and fire emissions (Rabin et al., 2017; Li et al., 2019; Hantson et al., 2020). However, this extensive processing chain requires fine-tuning many parameterizations and formulae, each of which has the potential to alter the outcome substantially. As a result, current state-of-the-art process-based fire models

are not always able to reproduce observed fire events (Burton et al., 2024; Park et al., 2024), and their projections contain substantial spread (Teckentrup et al., 2019; Lange et al., 2020; Thiery et al., 2021; Grant et al., 2025). Moreover, (sub)national fire databases are often incomplete and inconsistent (Bowman, 2018; Gincheva et al., 2024)

Machine learning algorithms have the advantage of being able to fit (non-linear) functions to data rather than prescribing them manually. In complex tasks, such as fire modelling, where the real world relations and interactions are hard or near-

impossible to pin down mathematically, machine learning can provide a valuable solution (Qi and Majda, 2020; Bracco et al., 2025). At the same time, machine learning often lacks interpretability (Rudin, 2019; Yang et al., 2024; Bracco et al., 2025), which can be a disadvantage compared to process-based models when process understanding or fine-grained control is the primary objective. Thus, machine learning can serve as a complementary rather than a substitutive approach to process-based fire modelling.

Here we present a data-driven fire model "BUrned area modelling by Recurrent Neural Networks (BuRNN)". BuRNN combines traditional fire model inputs and intermediary DGVM outputs such as Gross Primary Production (GPP) with machine





learning to predict burned area. We first describe the architecture and training process of the model. Then, we evaluate the skill of BuRNN against satellite data, using state-of-the-art process-based wildfire models as benchmark. Next, we attempt to understand the inner workings of BuRNN through XAI methods. Finally, we apply BuRNN to generate a monthly gridded burned area reconstruction from 1901 to 2019 at $0.5° \times 0.5°$ spatial resolution and evaluate this new dataset against regional wildfire records.

## 2 Materials & Methods

### 2.1 Data

To train BuRNN, we make use of six different data sources. BuRNN is trained on a monthly timescale and receives 27 features as input, each providing information on (i) climate, (ii) land or vegetation properties or (iii) socio-economic conditions (Table 1). Climate-related variables are: (i) monthly mean of the daily maximum temperature, mean monthly precipitation and mean monthly wind speed from the daily NOAA-CIRES-DOE 20th Century Reanalysis version 3 homogenized to W5E5 (20CRv3-W5E5) product (Compo et al., 2011; Slivinski et al., 2021; Lange, 2019; Lange et al., 2021), (ii) monthly mean Fire Weather Index (FWI) calculated from 20CRv3-W5E5, (iii) Standardised Precipitation-Evapotranspiration Index (SPEI) with a 1, 3 and 6 month time lag calculated through the FAO-56 Penman-Monteith estimation for potential evapotranspiration (Beguería et al., 2014) and (iv) lightning density. The land and vegetation characteristics are (i) land cover from the Community Land Model (CLM), which are generated based on Land Use Harmonization phase 2 (LUH2; Hurtt et al., 2020), (ii) land use provided by Inter-Sectoral Impact Model Intercomparison Project (ISIMIP) also based on LUH2 and (iii) intermediate DGVM outputs (ensemble mean) from the ISIMIP biome sector for GPP (n=7), Carbon Mass in Vegetation (cVeg) (n=3) and Leaf Area Index (LAI) (n=5) Table A1. Lastly, socio-economic conditions are provided by ISIMIP in terms of population densities and Gross Domestic Product (GDP) (Table 1). The LUH2 derived data from ISIMIP and CLM was linearly interpolated from a yearly to monthly timescale. Moreover, we removed and grouped a number of related land use/land cover classes in order to bring the total number of features down. Additionally, earlier research has pointed out that the driving factors for savannah fires are different on different continents (Lehmann et al., 2014; Alvarado et al., 2020; Simpson et al., 2022). Therefore, we also split the original C4 grasses variable into three additional variables based on their location i.e., (i) North-America and South-America (C4 grasses Americas), (ii) Africa (C4 grasses Africa) and (iii) Europe, Asia and Oceania (C4 grasses Eurasia_Oceania). We chose these input variables as all are available on a monthly timescale from 1901 onwards at a $0.5° \times 0.5°$ spatial resolution (or higher) and represent many drivers, or proxies thereof, of fire behaviour. To train BuRNN, we use GFED5 as target data (Chen et al., 2023b), we remapped the original $0.25° \times 0.25°$ grid to $0.5° \times 0.5°$ using area-weighted regridding from the Python Package *Iris - SciTools*. GFED5 derives burned area estimates for 2001–2020 from the Moderate Resolution Imaging Spectroradiometer (MODIS) MCD64A1 product (Giglio et al., 2018), applying region-, land cover-, and tree cover-specific corrections for commission and omission errors based on spatiotemporally aligned Landsat and Sentinel-2 burned area observations. Burned area in croplands, peatlands, and deforestation regions is separately estimated using MODIS active fire detections (Giglio et al., 2016). To extend the record back to 1997, active fire data from the Along-Track Scanning





Radiometer (ATSR) and the Visible and Infrared Scanner (VIRS) were used, which carry higher uncertainties (Chen et al.,
2023b). Although GFED5 almost doubles the observed burned area compared to other satellite products, we consider it most
suitable for ground truth as it matches high-resolution burned area observations for Africa (Chuvieco et al., 2022). Moreover,
literature suggests that 'traditional' burned area products, such as FireCCI51 severely underestimate actual burned area (Zhu
et al., 2017; Franquesa et al., 2022; Khairoun et al., 2024), supporting our choice for GFED5 as target dataset.

**Table 1.** List of the 27 features provided to BuRNN along with their origin.

| Type | Source | Description | Number of Features |
|---|---|---|---|
| Climate | 20CRv3-W5E5 | We aggregate the daily values for daily maximum temperature (tasmax; in K), total precipitation (pr; in kg m$^{-2}$ s$^{-1}$) and near-surface wind speed (sfcWind; in m s$^{-1}$) to monthly means. | 3 |
| | | Canadian FWI calculated on a daily timescale from tasmax, pr, tasmax and near-surface relative humidity (hurs; in %) (van Wagner, 1987). These daily values are then aggregated to monthly means through CDO. | 1 |
| | SPEIbase | 1, 3 and 6-month lagged SPEI from Beguería et al. (2014) with potential evapotranspiration calculated via FAO-56 Penman-Monteith estimation. | 3 |
| | HistLight & WGLC | Lightning density provided by combining HistLight (1901-2009) and WGLC (2010-2019) (Kaplan and Lau, 2022a, b). | 1 |
| Land & vegetation | CLM | Land cover maps originating from LUH2 (Hurtt et al., 2020) and processed for use as input to the Community Land Model (CLM, Lawrence and Chase, 2007; Lawrence et al., 2019). We regrouped the original 17 land cover types into 11 groups (all represented as fraction of grid cell area): Urban, Lake, Crop, Bare Ground, Needleleaf tree, Broadleaf evergreen tree, Broadleaf deciduous tree, Broadleaf shrub - temperate, Broadleaf deciduous shrub - boreal, C3 grass and C4 grass. Then we added three additional variables for C4 grasses i.e., C4 grasses Americas, C4 grasses Africa and C4 grasses Europe-Asia-Oceania. | 14 |
| | ISIMIP | Land use maps originating from LUH2 and processed for use in ISIMIP (Volkholz and Ostberg, 2022). Given the similarity between the land cover and land use datasets, only the grid cell fractions managed pastures and rangeland were added to the feature list. | 2 |
| Socio-economic | ISIMIP | Rural and urban population along with GDP from ISIMIP3a (Volkholz et al., 2024; Sauer et al., 2024) . | 3 |

## 2.2   Model Description

We aim to design a machine learning model that is able to learn the lagged and cumulative effects of climate variability, land
use and socio-economic conditions on fire dynamics. Unlike traditional machine learning algorithms, which often treat each
observation independently, Long Short-Term Memorys (LSTMs) are capable to capture non-linear temporal dependencies in
sequential data (Hochreiter and Schmidhuber, 1997), making them ideal for our use case. Although LSTMs were originally



designed for natural language processing (Gers et al., 2000), LSTMs have also successfully been applied in a number of climate related applications such as modelling vegetation dynamics (Reddy and Prasad, 2018), predicting river streamflow (Hunt et al., 2022), weather forecasting (Karevan and Suykens, 2020) and even detection of forest fires (Cao et al., 2019). Therefore, we chose the LSTM as main component of BuRNN. The LSTM maintains its own hidden states acting as *memory*, which is updated dynamically in interaction with the input features. The hidden state at each time step is mapped to a single output using a dense neural layer, yielding the predicted burnt area fraction. Despite the simple model architecture, a couple of hyperparameters have to be chosen. To automate the search for optimal hyperparameters, we used the *Optuna* framework (Akiba et al., 2019). We used the Tree-structured Parzen Estimator (TPE) sampler inside the framework to find appropriate values for the learning rate, number of LSTM layers, hidden size of the LSTM layer(s), activation functions, number of dense neural layers, size of the dense neural layers and dropout fraction (Bergstra et al., 2011). Currently, BuRNN is a single layered LSTM with a hidden size of 64 connected to a dense neural layer with Rectified Linear Unit (ReLU) activation function given in Eq. 1 (see also Fig. 1):

$$\mathrm{ReLU}(x) = \max(0, x) \tag{1}$$

Given the nature of our data, our input variables (and targets) contain a high degree of spatial autocorrelation. Applying a traditional random train-test split or random train-test folds would likely lead to an overestimation of performance and poor predictive power (Diniz-Filho et al., 2008; Le Rest et al., 2014; Meyer et al., 2019). Therefore, we trained our LSTM networks according to a region-based cross-validation. We split our data according to 43 Intergovernmental Panel on Climate Change (IPCC) land regions (we removed the two Antarctic regions and Greenland) and manually grouped these regions into 11 folds (Fig. A1), whereby we made sure that the 3-4 regions in each fold represent different continents and biomes (Iturbide et al., 2020). For each fold, we use two different folds as validation set and the remaining 8 folds as training set. We repeat this five times for each fold, each time with two different folds as validation set. For example, when fold 1 is chosen as test fold, we first select folds 2 and 3 as validation set and folds 4-11 as training set. Then we choose folds 4 and 5 as validation set and folds 2-3 and 6-11 as training set, etc. This results in a total of 55 (11 times 5) models. Then, when we make predictions with our model for an IPCC region, it is the mean estimate of five LSTMs which have never seen data for that IPCC region before. The validation folds are used to monitor model convergence and overfitting by using the early stopping algorithm; As soon as model performance on these validation folds started to decrease after a given set of training iterations, training was stopped and the best model was restored. This model was then used to make predictions on the independent test set.

Before training, we combine the data from all different sources, convert the time dimension to have identical units and split them into the 11 pre-defined folds. We normalize the training data and use the mean and standard deviation of the training set to normalize the validation folds (and the test fold during prediction). Each time we change the training folds, we undo the normalization operation, and redo it based on the mean and standard deviation of the new training set. Additionally, we log-transform the target variable (GFED5 percentage burned area) as the original data is strongly right-skewed. Pre-processing





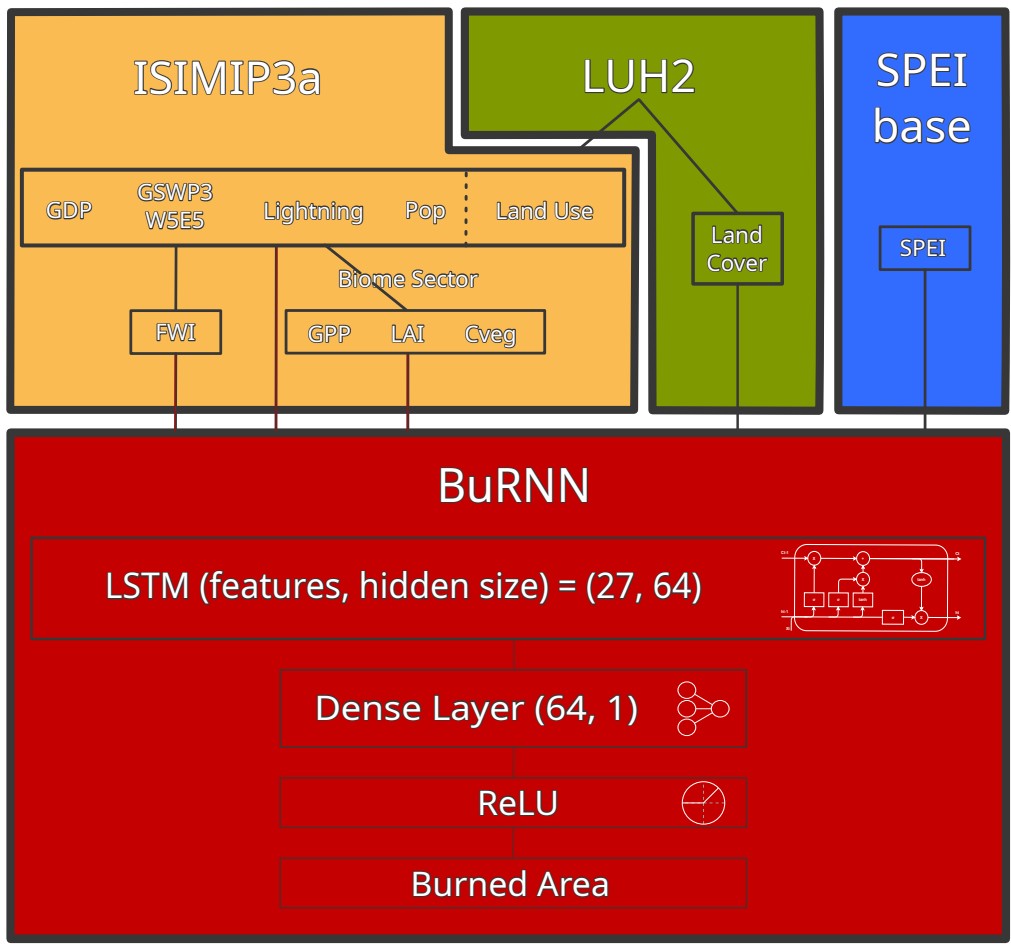

**Figure 1.** Structure of BuRNN. The top row denotes the origin of all the features supplied to our model, split into the three main sources (ISIMIP3, LUH2, and SPEIbase). The red rectangle reflects the architecture of BuRNN.

of our data happens through a combination of *Xarray* and *NumPy*. We use *PyTorch* and *PyTorch Lightning* to build our model architecture and to handle training and validation (Paszke et al., 2019; Falcon and The PyTorch Lightning team, 2019). In

the training phase the LSTM layer is followed by batch normalization. During training, we provide the samples in batches of 32, an error is calculated based on the cumulative error of the predictions for these 32 samples (see further) after which the model is updated/improved. Batch normalization normalizes the features of each batch (based on the batch's mean and standard deviation) and results in faster and more stable training (Santurkar et al., 2018). This layer is followed by a dropout layer for which the optimal dropout fraction was found to be 0.2. This randomly ignores, on average, 20% of the connections

between the LSTM and dense layer, which has been proposed to improve generalisation and reduce overfitting (Srivastava et al., 2014). During training we ignore the first 36 predictions (3 years) to allow the LSTM's memory state to spin up and then




evaluate the predictions of the following 3 years using Mean Squared Error (MSE) as loss function. In each epoch, we pass each pixel/location in the training set once and randomly select a 6 year time slice.

## 2.3 Model Evaluation

We evaluate our predictions for 1997-2019, the common period between the observational GFED5 product and the ISIMIP fire sector simulations (forced with the GSWP3-W5E5 reanalysis). We evaluate our 3D (time, latitude, longitude) data cubes for several metrics in different dimensions (spatial, temporal and spatio-temporal). By calculating the Root Mean Squared Error (RMSE) between the modelled and observed 3D cubes, we obtain an error expressed in % burned area. Similarly, by calculating the Pearson correlation we obtain a metric that informs on spatial and temporal patterns, ignoring the mean and scale

bias the process-based models and BuRNN have (Hantson et al., 2020; Burton et al., 2024). The spatial pattern is evaluated by computing the mean over time, resulting in a 2D data cube (latitude, longitude), and we calculate both spatial RMSE and correlation. Similarly, by taking the sum over the spatial domain (latitude and longitude), we arrive at a monthly and yearly time series of global burned area. We calculate yearly correlation, which assesses the interannual variability, and monthly correlation, which represents seasonality.

## 155 2.4 Driver Analysis

To better understand the inner workings of BuRNN, which is *in se* a black box model, we employ an explainable AI method. SHapley Additive exPlanations (SHAP) is a method from game theory that investigates the effects features have on the prediction outcomes of machine learning models (Lundberg and Lee, 2017). SHAP assigns each feature a score that represents its contribution to the difference between the actual prediction and the average prediction over a dataset. It provides a consistent

way to quantify how much each feature pushes the prediction higher or lower. We randomly select 50 samples for each GFED region for each model and calculate SHAP values over 24 months (see Section 2.2). We use GradientExplainer from the aptly named SHAP Python package to calculate the SHAP values. GradientExplainer calculates the gradients of the input features (how much does the output of the model change when each input is changed slightly) and approximates the SHAP values by integrating the gradients over the input features, moving from the average input (mean of each feature in the dataset) to

an actual sample in the dataset. The GradientExplainer approach is an extension of integrated gradients proposed originally in Sundararajan et al. (2017). We note here upfront that SHAP does not provide causal insights into the real-world processes underlying the data. Rather, SHAP values offer a post hoc explanation of the model's internal logic by attributing contributions to input features in a way that reflects the model's learned associations. When applied to structured or interdependent data, SHAP values can be particularly difficult to interpret and may not yield a complete picture of how the model uses such inputs.

Nonetheless, they can still offer a useful high-level perspective on the patterns and dependencies the model has captured. This analysis therefore serves the purpose of understanding the statistical associations the model has learned, rather than uncovering mechanistic relationships inherent to the system under study.



## 2.5 Burned Area Reconstruction

After training, we use the models to simulate burned area for the period 1901-2019. During training we employed a 3 year
spinup period (see Section 2.2). Therefore, we add 1901-1903 in front of the dataset so this can be used as spinup. We analyse
this full reconstruction per region and also compare it against a 1997-2019 run to verify the stability of the model.

Moreover, we compare the reconstruction to the FireCCiLT11 product, which is based on Advanced Very-High-Resolution
Radiometer (AVHRR; Otón et al., 2021). FireCCiLT11 is available from 1982-2018, with the exception of 1994. We calculate
the regional 1982-1993 correlations for annual burned area between BuRNN and FireCCiLT11 and compare those to the 1997-
180 2018 correlations between BuRNN and FireCCiLT11 and between GFED5 and FireCCiLT11. Ideally, the latter values are high,
indicating both observational products are in agreement. If this is the case, then a good reconstruction (1982-1993) should have
a similar correlation (to FireCCiLT11) for both periods.

Additionally, we compare the reconstruction to regional datasets where available (see subsection 3.3). For Canada, we assess
the National Burned Area Composite (NBAC) and National Fire Database (NFDB) datasets. NBAC is fire polygon database
from Landsat (30m) starting from 1972 and contains data on ∼35000 fires (Canadian Forest Service, 2024). NFDB combines
data from various Canadian agencies and contains data for over 700 000 fires between 1959 and 2022 (Hanes et al., 2019). For
the Unites States, we compare our reconstruction to Monitoring Trends in Burn Severity (MTBS) and Fire Occurrence Database
(FOD). MTBS estimates burned area from Landsat and provides data on fires $>2km^2$ since 1984 (Picotte et al., 2020). FOD
encompasses fire records from several US agencies for 1992 to 2020 and excludes prescribed burning (Short, 2022). For Brazil
we use data from the MapBiomas project, which produces gridded burned area over Brazil from 1985 to 2023 based on Landsat
(Souza Jr et al., 2020). For Chile, the database is managed by Chilean Forest Service (CONAF) and is also based on Landsat,
it contains information on over 200 000 fires from 1985 to 2021. We obtained the Chilean data from Gincheva et al. (2024).
European Forest Fire Information System (EFFIS) provides us with country-level data on non-agricultural fires for 21 countries
in the EU (excluding Austria, Belgium, Denmark, Ireland, Luxembourg and Malta). The data comes from the individual EU
countries and is available for different time periods for each country, the earliest is 1980 for Portugal. Lastly, we also asses
fires over Australia, making use of data from over 75% of the Australian surface area. Data was provided by different state and
territory agencies and was combined by Gincheva et al. (2024) and is available from 1950 to 2021. All these datasets come
with a number of caveats, especially in the earlier periods. They are (i) often incomplete, (ii) use different protocols between
products, but also for different time periods within a dataset and (iii) they report different things (some exclude agricultural
and/or managed fires, others exclude small fires) (Gincheva et al., 2024). Nonetheless, they are the best independent reference
data we have available.



## 3 Results

### 3.1 Model Evaluation

Our global-scale evaluation results highlight that BuRNN outperforms all process-based fire models on each of the skill met-
rics we consider (see Section 2.3, except for interannual variability, where only CLASSIC outperforms BuRNN; Table 2).
BuRNN has a RMSE of 1.67, while the process-based fire models fall between 2.02 and 3.07. Similarly, the correlation fac-
tor is 0.7 and between 0.01 and 0.51 for the FireMIP models. The spatial RMSE of BuRNN is 0.5, while the process-based
models fall between 0.82 and 1.32. The spatial correlation is 0.89 for BuRNN and between -0.01 and 0.69 for the FireMIP
models. Monthly correlation, representing seasonality, is 0.81 for BuRNN and between -0.13 and 0.66 for the FireMIP models.
BuRNN's yearly correlation, representing interannual variability, is 0.72, while it is between -0.34 and 0.80 for the process-
based models (Table 2). Three example maps of burned area prediction by BuRNN are shown alongside those of GFED5 and
the two best-performing process-based models in Figs. A4 to A6. Hence, with one exception (yearly correlation of the CLAS-
SIC model) BuRNN scores better than any other fire model for each considered performance metric (total of 54 model-metric
combinations). These evaluation results thus overall indicate that at the global scale, BuRNN largely outperforms state-of-the-
art global wildfire models. Fig. 2 depicts the mean monthly burned area from GFED5 (upper left), BuRNN (upper right) and
the nine FireMIP models. In general, the spatial pattern of BuRNN matches closely the pattern of GFED5. However, there is a
clear underestimation of burned area in the Northern Australian savannahs and mainland Southeast Asia. This is made further
clear in Fig. 3, which shows the difference in mean monthly burned area between BuRNN and GFED5 (upper right) and be-
tween the FireMIP models between the FireMIP models and GFED5. The density plot in the upper left depicts the distribution
of the error over all land pixels for BuRNN and the FireMIP models, where the difference between GFED5 and each of the
models is considered the error. The distribution of BuRNN falls more closely around zero than any of the FireMIP models,
indicating again better spatial performance.

We also evaluate our results across 14 fire regions defined by Giglio et al. (2010) in Fig. 4. We find a general tendency
to underestimate the annual burned area in most regions. Interannual variability is relatively well modelled, although the
amplitude is lower than observed for most regions. However, there are differences in performance across regions. Regions
such as Temperate North America (TENA), Northern Hemishphere South America (NHSA) and Equatorial Asia (EQAS) are
excellently modelled by BuRNN. In the majority of the regions, BuRNN captures the pattern of the interannual variability well,
but consistently underestimates the amplitude and total burned area, for instance in Boreal North America (BONA), Central
America (CEAM), Southern Hemishphere South America (SHSA), Europe (EURO), Northern Hemishphere Africa (NHAF),
Southern Hemisphere Africa (SHAF) and Southeast Asia (SEAS). In the Middle East (MIDE), BuRNN simulates the mean
annual burned area well, while the interannual variability is off especially for the later years. In Boreal Asia (BOAS), our
model simulates too little burned area, which is likely due to having a similar environmental setting as BONA where annual
burned area is much lower. In Central Asia (CEAS), our simulations do not match the observed interannual variability or mean
annual burned area. In Australia and New Zealand (AUST), total burned area is much lower than observed, while interannual
variability is reasonable in terms of general pattern (which years have high or low burned area), but again too low in amplitude.



**Table 2.** Global evaluation scores of BuRNN and the FireMIP models. Colour scaling has been done based on the normalized values (value - row mean)/(row standard deviation) with the minimum and maximum values set to -2 and 2, respectively. Better scores (lower for RMSE and higher for Pearson correlation) are marked in blue, while worse performance is in red.

| | BuRNN | CLASSIC | ELM-ECA | JULES-INFERNO-VN6P3 | LPJ-GUESS-SIMFIRE-BLAZE | LPJ-GUESS-SPITFIRE | LPJmL5-7-10-fire | ORCHIDEE-MICT | SSiB4-TRIFFID-Fire | VISIT |
|---|---|---|---|---|---|---|---|---|---|---|
| RMSE | 1.67 | 2.18 | 3.07 | 2.16 | 2.30 | 2.32 | 2.37 | 2.83 | 2.02 | 2.56 |
| Spatial RMSE | 0.50 | 0.87 | 1.02 | 0.82 | 0.85 | 0.96 | 0.89 | 1.24 | 0.75 | 1.32 |
| Correlation | 0.70 | 0.40 | 0.11 | 0.37 | 0.29 | 0.26 | 0.30 | 0.18 | 0.51 | 0.01 |
| Spatial Correlation | 0.89 | 0.61 | 0.32 | 0.62 | 0.58 | 0.43 | 0.53 | 0.40 | 0.69 | -0.01 |
| Monthly Correlation | 0.81 | 0.55 | 0.11 | 0.39 | 0.49 | 0.18 | 0.35 | -0.13 | 0.66 | 0.11 |
| Yearly Correlation | 0.72 | 0.80 | 0.07 | 0.29 | 0.57 | 0.38 | -0.01 | -0.34 | 0.69 | -0.21 |

good                                            average                                            bad

Global annual burned area is mostly dominated by the (African) savannah regions; at this scale the discrepancy between observed and simulated global burned area is therefore mainly due to the underestimation of BuRNN in NHAF, SHAF and AUST. Nonetheless, in most regions BuRNN again outperforms the process-based fire models over most metrics (Table 3).

Next, we repeat this evaluation procedure using the 2001-2019 FireCCI51 observational dataset as reference. We do this because our model is specifically trained to predict GFED5 burned area, while the process-based models are not. Although the absolute values between FireCCI51 and GFED5 differ, a similar pattern as Table 3 is observed when comparing BuRNN and the process-based models against FireCCI51, that is, BuRNN outperforms the process-based models in most regions and for most metrics (Table A17).

RMSE metrics vary in magnitude across different regions as they have different total burned areas. However, also the correlation metrics show large inter-regional differences. For example, in MIDE BuRNN outperforms all process-based models with a yearly correlation of 0.42. In EQAS, six out of the nine fire models outperform BuRNN in yearly correlation, but BuRNN still has a high absolute score of 0.89 in this region. In general, the higher the average monthly burned area in a region, the better/easier predictions are for that region. This partially explains why regions with high burned areas, such as NHAF and





**Figure 2.** Mean monthly burned area over 1997-2019 for the GFED5 satellite product, BuRNN and the nine FireMIP models.





**Figure 3.** Spatial difference in mean monthly burned area (over the period 1997-2019) between GFED5 observations and the model simulations (including BuRNN). The left upper panel shows the distribution of pixel values per model, the more closely centered around 0, the better the modelled burned area pattern.





**Table 3.** Regional evaluation scores of BuRNN. Colour scaling has been done based on the ranked values compared to the nine process-based fire models, with the minimum RMSE and maximum correlations coloured blue (best) and the highest RMSE and lowest correlation coloured red (worst).

| | AUST | BOAS | BONA | CEAM | CEAS | EQAS | EURO | MIDE | NHAF | NHSA | SEAS | SHAF | SHSA | TENA |
|---|---|---|---|---|---|---|---|---|---|---|---|---|---|---|
| RMSE | 2.67 | 1.20 | 0.41 | 1.25 | 1.16 | 0.64 | 0.40 | 0.30 | 3.51 | 0.68 | 2.55 | 3.43 | 0.88 | 0.38 |
| Spatial RMSE | 0.96 | 0.30 | 0.06 | 0.37 | 0.27 | 0.20 | 0.11 | 0.11 | 1.01 | 0.25 | 0.88 | 1.05 | 0.29 | 0.10 |
| Correlation | 0.33 | 0.43 | 0.18 | 0.52 | 0.35 | 0.37 | 0.32 | 0.19 | 0.77 | 0.63 | 0.59 | 0.74 | 0.53 | 0.15 |
| Spatial Correlation | 0.57 | 0.69 | 0.49 | 0.63 | 0.64 | 0.47 | 0.63 | 0.36 | 0.93 | 0.81 | 0.64 | 0.89 | 0.67 | 0.38 |
| Monthly Correlation | 0.84 | 0.90 | 0.55 | 0.83 | 0.76 | 0.88 | 0.75 | 0.79 | 0.97 | 0.93 | 0.94 | 0.99 | 0.95 | 0.73 |
| Yearly Correlation | 0.91 | 0.63 | 0.74 | 0.79 | 0.70 | 0.89 | 0.50 | 0.42 | 0.74 | 0.73 | 0.54 | 0.63 | 0.74 | 0.66 |

best                  average                  worst

SHAF have high monthly correlations as opposed to BONA and TENA where monthly correlations between modelled and
observed burned area are lower. Similar observations can be made over the spatial correlation, where NHAF and SHAF are
again the regions with the best modelled spatial burned area and BONA and TENA the two regions where the spatial pattern
is least well modelled of all regions, although the difference in correlation between best and worst region is smaller than with
monthly correlations. The likely reason for the lower spatial correlation (both for BuRNN and the process-based models) in
these regions is the stochastic nature of fires on these spatial and temporal scales. For example, large regions (many pixels)
of Canadian forest are quasi-identical in terms of how their monthly input features look like. In these regions large fires are
associated with periods of high fire weather danger, which usually occurs over many pixels on this scale. However, when a
large fire event happens only a few pixels will see very high burned areas, where exactly these will occur is difficult to predict.
Therefore, BuRNN and many process-based models do not predict these large fires in specific pixels but spread out the burned
area over a larger area. This in turn leads to lower spatial predictive power in these regions. Moreover, in a number of regions
interannual variability is poorly modelled. Earlier research has shown that interannual variability in e.g., BONA, TENA, MIDE,
BOAS and AUST is usually related to climate-related factors (Chuvieco et al., 2021). As BuRNN underestimates interannual
variability in many of these regions, a possible improvement would involve including more climatic variables, such as vapour
pressure deficit and evapotranspiration, during model training.





**Figure 4.** Annual sums of regional burned area by BuRNN (orange) and the GFED5 satellite observations (blue) for 1997-2019.





## 3.2 Drivers of BuRNN

We find that the FWI and daily maximum temperature (tasmax), the presence or absence of bare ground and C4 grasses, and GPP are the most impactful features (Fig. 5) across most regions. This suggests that although the Canadian FWI was originally designed to be used in Canadian forests, it can provide relevant information for many, if not all, regions in the world. However, regional differences in SHAP values can be observed. For example, grassland indicators show up in regions with considerable grassland fractions e.g., SHAF, SHSA and AUST, but in EURO, TENA and BOAS the absence of African C4 grasses also

shows up as a reducing factor i.e., because there are no African C4 grasses there, which it cannot have by default, burned area is lower than the global average. From the perspective of BuRNN this makes sense as the presence of African C4 grasses are often an indicator of high burned areas, so an absence is likely associated with lower than average burned area. We also note that NHSA, SHSA, NHAF, SHAF and AUST, which contain notable savannah regions, are the main regions where GPP is ranked higher than in most regions. This corroborates earlier assessments that state that savannah fires are predominantly

limited by fuel and moisture availability (Lehmann et al., 2014; Alvarado et al., 2020; Takacs et al., 2021). In MIDE and CEAS, both regions with large unvegetated deserts, bare ground is the most impactful feature. This can be explained by a high value for bare ground negating the relevance of all other features in a pixel. Additionally, bare ground is also among the top five predictors in (i) BONA and BOAS, which have notable regions without or with little vegetation in the upper North, (ii) TENA, NHAF, SEAS and AUST, which have desert areas. In SEAS, the rural population is the most important predictor and it is the

only region where this predictor even makes the top five.

## 3.3 Application: a burned area reconstruction for the $20^{th}$ Century

Fig. 6 shows the global and regional annual burned area as modelled by BuRNN for the period 1901-2019. BuRNN simulates that globally, from 1901-2000 there has been a slight increase in burned area, which is mainly attributed to an increase in burned area in SHAF in that same period. Notable changes are found in TENA, NHSA EURO, MIDE and SHAF. In TENA

BuRNN simulates an increasing trend in burned area from 1901 until ∼1955 after which a sharp decline is observed from ∼1960 until ∼1990. In NHSA a small but consistent increase in burned area is modelled for the entire 1901-2019 period. In EURO a first period of high burned area with large interannual variability is modelled from 1901 until ∼1950, after which a stark declining trend is modelled by BuRNN. the latter, more recent declining trend is also observed in the EFFIS database. In MIDE burned area starts low from 1901 until ∼1930, after which is quickly increases until ∼1950, after which a slight negative

trend is simulated. Lastly, for SHAF a positive burned area trend is modelled for the 1901-2010 period, after which burned area again decreases in the last ∼10 years. Next, we also want to compare the 1982-1997 part of the BuRNN reconstruction to FireCCiLT11. Fig. A2 shows the 1982-2018 regional annual burned area from BuRNN and FireCCiLT11. The annual burned area correlations for 1982-1993 and 1997-2018 between FireCCiLT11 and BuRNN are listed in Table A16 along with the 1997-2018 annual burned area correlations between GFED5 and FireCCiLT11. The annual correlation between the two products is

relatively low. However, the uncertainty in burnt area estimates for this period is relatively high, and on average the correlation





**Figure 5.** SHAP values of BuRNN per region, indicating which features are most important in each region. SHAP values indicate the importance of a feature in affecting the prediction of a model. Blue values show the effect of low values of the predictor on the prediction, while pink values indicate the effect of high values of the predictor. Features are ranked from top to bottom in terms of total importance, the top five are shown for each region.



between BuRNN and FireCCiLT11 for the early period is higher than between the two observational products themselves for 1997-2018 Table A16.

Additionally, we compare our reconstruction to regionally available burned area databases. Fig. 7 shows the burned area from EFFIS reported by 21 countries in the EU. Both correlation and bias between this EFFIS database and BuRNN is generally

high, with BuRNN simulating higher burned areas than EFFIS. We note however that the reported burned area by EFFIS does not include cropland fires, as opposed to BuRNN, explaining the high absolute bias as cropland fires account for ∼two-thirds to three quarters of the total burned area in Europe (Chen et al., 2023b). In Fig. A3, a further comparison is made for 5 more regions (Canada, US, Brazil, Chile and Australia) where the correlation between national databases and BuRNN is only high in Brazil. The likely explanation for this discrepancy lies in the data collection. Correlation between BuRNN and EFFIS is

high for individual countries, but is close to 0 when assessed over the 21 European countries combined for the entire period. As each national dataset inside the EFFIS database has a different start and end date, it makes calculating interannual variability inconsistent (unless we restrict the database to only those years available in all countries, which is 2017-2019). Similarly, many of the other national databases, like those in Canada, US and Australia, are composed of regional data sources that come available in different time periods mixed in with satellite images (usually LandSat) when available. In contrast, MapBiomas

in Brazil has a single data source (LandSat) and thus does not suffer from this, there correlation with BuRNN is high (0.78). Therefore, we believe BuRNN shows a good correlation with these independent data sources whenever the data sources have consistent reporting of burned area. Moreover, in Europe a decreasing trend in annual burned area has been reported, especially in the Mediterranean (Rodrigues et al., 2013; Turco et al., 2016; Chen et al., 2023b). This is in line with the reconstruction of BuRNN.





**Figure 6.** BuRNN's simulation of total annual burned burned area from 1901 to 2019 (orange) for each of the fourteen GFED5 regions and globally.



**Figure 7.** Annual sums of national burned area by BuRNN (orange) and as reported by EFFIS. Methods for collecting and reporting burned area differ by country (and may differ throughout time). Note that EFFIS does not include cropland fires.





## 4   Discussion

Scientific performance aside, BuRNN has a second benefit compared to process-based models i.e., speed and cost of running the model. Running the full 1901-2019 reconstruction (for all the 55 models) takes approximately an hour in total on a single CPU core on our HPC cluster. This is in stark contrast to the computational cost required to run fire-coupled DGVMs, which require hundreds up to tens of thousands of CPU hours. Of course, the major cost of running BuRNN is in the training phase, which typically takes around 10 hours on a single GPU (NVIDIA GeForce 1080Ti). Although the speed and performance of this first version of BuRNN are excellent, it does come at the expense of interpretability. As with most deep learning architectures, BuRNN does not physically relate drivers to responses. We have done effort to alleviate this through our analysis of SHAP values, which approximates feature importance, but this understanding is not on par with our knowledge of the mechanisms in process-based models. Conversely, data-driven models can potentially contribute to improved process understanding: if we can unravel why and how BuRNN outperforms these process-based fire models, we can leverage that knowledge to improve the process-based models.

During training, we explicitly aimed to prevent overfitting and maximize generalisability in several ways. We employed a region-based cross validation to counteract the high spatial autocorrelation in our data, we used early stopping, applied normalization during preprocessing on the training data, batch normalization after the LSTM layer and dropout after the linear layer. We subsequently evaluated BuRNN in multiple ways over a number of metrics against multiple products. First, we evaluated the performance of BuRNN by assessing its error scores to GFED5, taking into account that for any region in the world, BuRNN has never seen data from that region before. Then we calculated spatiotemporal, spatial and temporal error scores and correlation of BuRNN to GFED5 and FireCCI51. We repeated this for the process-based fire models participating in ISIMIP3a and compared the relative performances, showing that in most regions over most metrics BuRNN outperforms state-of-the-art fire models. Our burned area reconstruction holds major promise for assessing spatial fire patterns in the pre-satellite era. To assess its quality, we compared our 1982-1993 reconstruction to the FireCCiLT11 remote sensing product and national census data. However, the low correlation between GFED5 and FireCCiLT11, highlights important observational uncertainty in the early satellite record, calling for caution when interpreting our AI-based reconstruction relative to FireCCiLT11 in this period. By comparing our reconstruction of BuRNN to national databases wherever available, we can potentially obtain a sense of regional product quality. We find particularly good correlation with national databases in the EU and Brazil. Databases from Canada, US, Chile and Australia showed poor correlation to the BuRNN reconstruction, likely caused by the heterogenous nature of these reference datasets.

## 5   Conclusions

Compared to process-based fire models, BuRNN pushes the state of the art in terms of simulation quality of burned area, demonstrating the potential for machine learning to improve the predictive capabilities in regional-to-global scale fire modelling. As fire behaviour is expected to have changed and continue to change due to climate change, understanding how they have evolved and will evolve is important for understanding our ecosystems, emissions and land use changes. BuRNN



substantially improves our capabilities for simulating fire behaviour in all regions of the world compared to state-of-the-art process-based fire models. However, as a machine learning model its interpretability remains below that of conventional fire

models. To address this limitation, we applied XAI to unravel some of the inner workings of BuRNN. From this, we conclude that in most regions, BuRNN prioritizes features that are relevant for that region. This includes, for example, bare ground in regions with deserts and C4 grasses in all savannah regions. As an application, we apply BuRNN to reconstruct global monthly burned area at 0.5° x 0.5° spatial resolution over the period 1901-2019. While a valuable dataset for studying historical burned area patterns, it is a challenge to assess the quality of the product, given considerable discrepancy between different satellite-

based burned area products and between the satellite products and national inventories. As the effects of climate change and socio-economic drivers on fire behaviour are largely unknown (quantitavely), BuRNN can aid in better unravelling past burned area patterns, which can improve carbon cycle modelling, help fire risk prevention and inform policy makers.





**Figure A1.** Division of the 43 regions into 11 folds, used for training the models.





**Figure A2.** Annual sums of regional burned area by BuRNN (orange) and the FireCCiLT11 observations (blue) for 1982-2018.



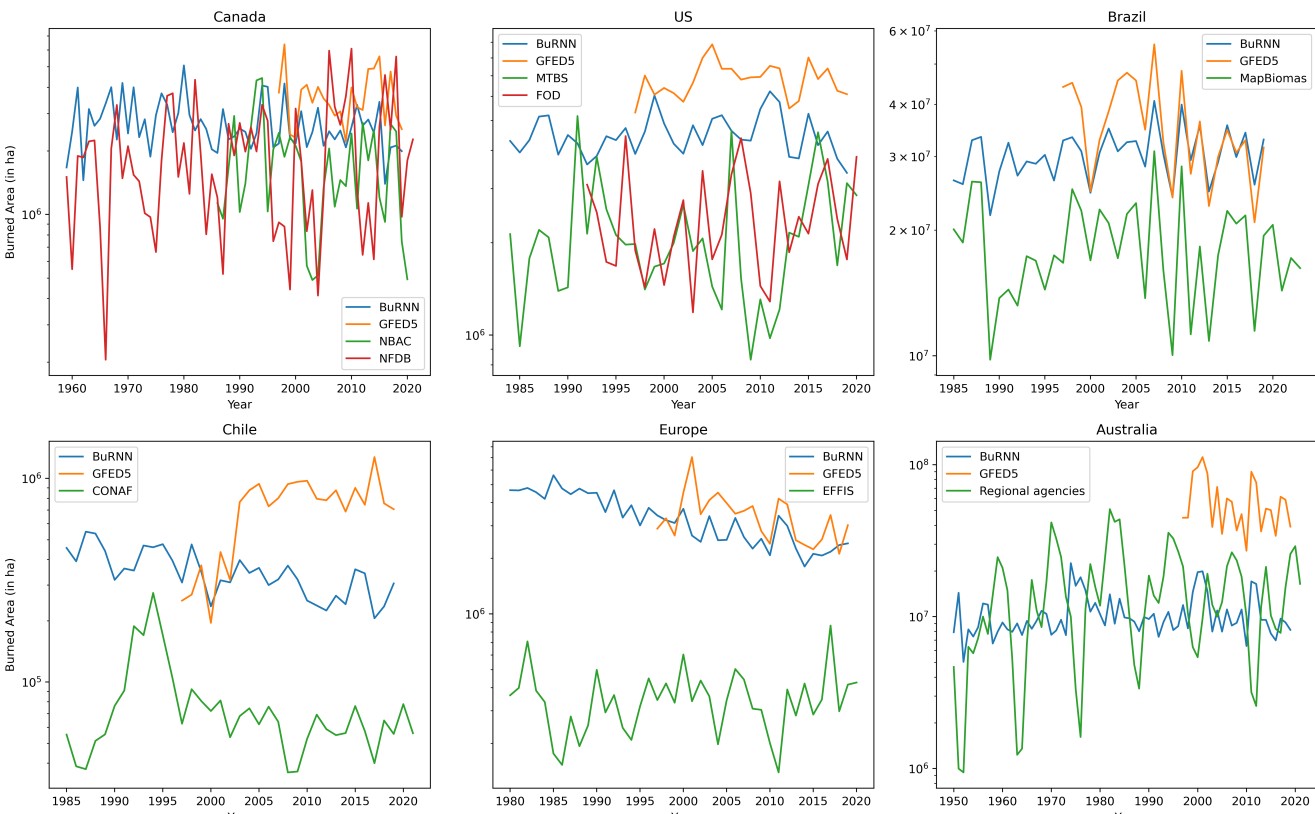

**Figure A3.** Comparison of BuRNN to regional burned area databases. Note that in some regions managed and/or agricultural fires are not reported.



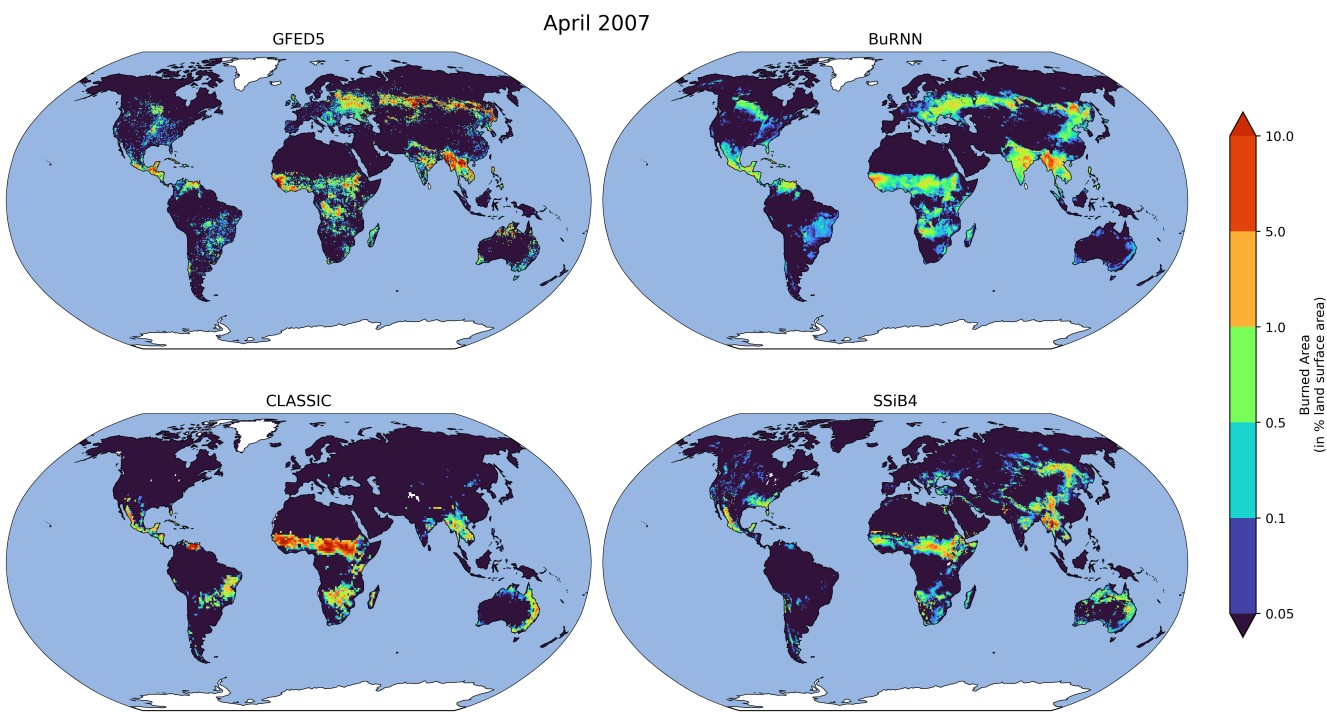

**Figure A4.** Comparison of BuRNN to GFED5 along with two process-based models (SSiB4 and CLASSIC) for April 2007.



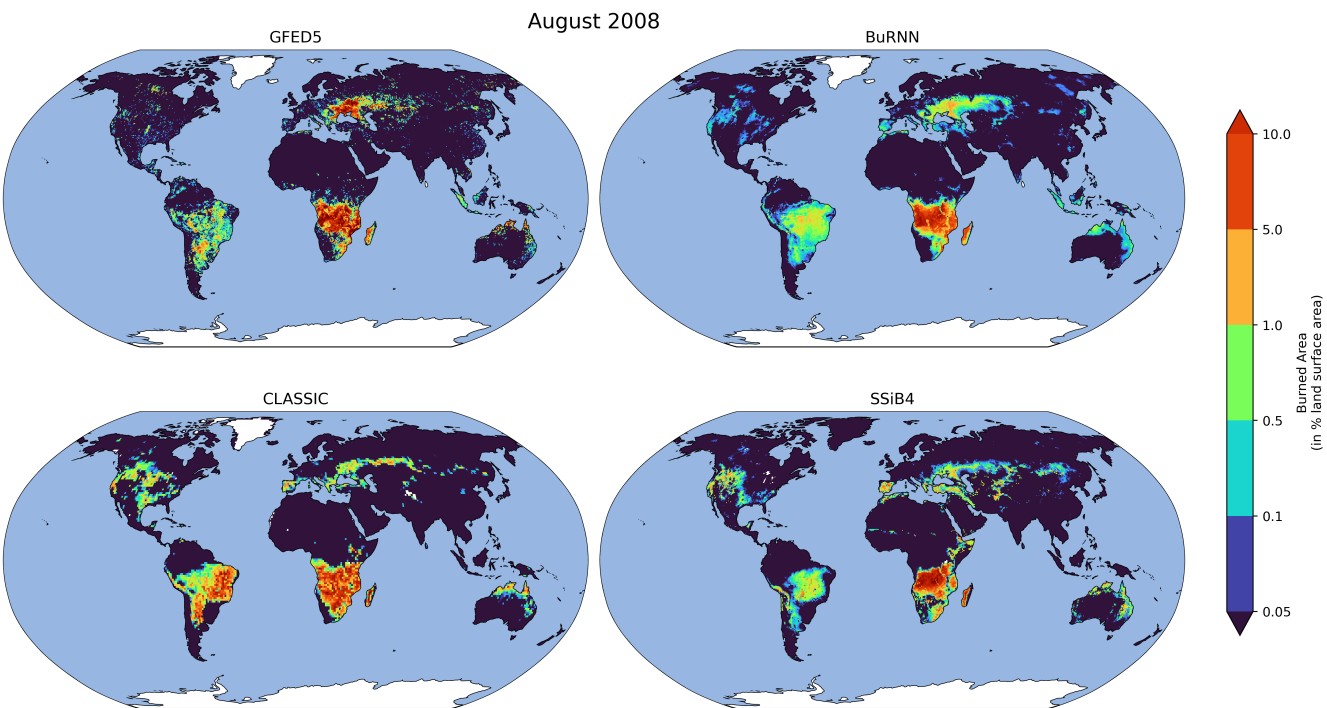

**Figure A5.** Comparison of BuRNN to GFED5 along with two process-based models (SSiB4 and CLASSIC) for August 2008.



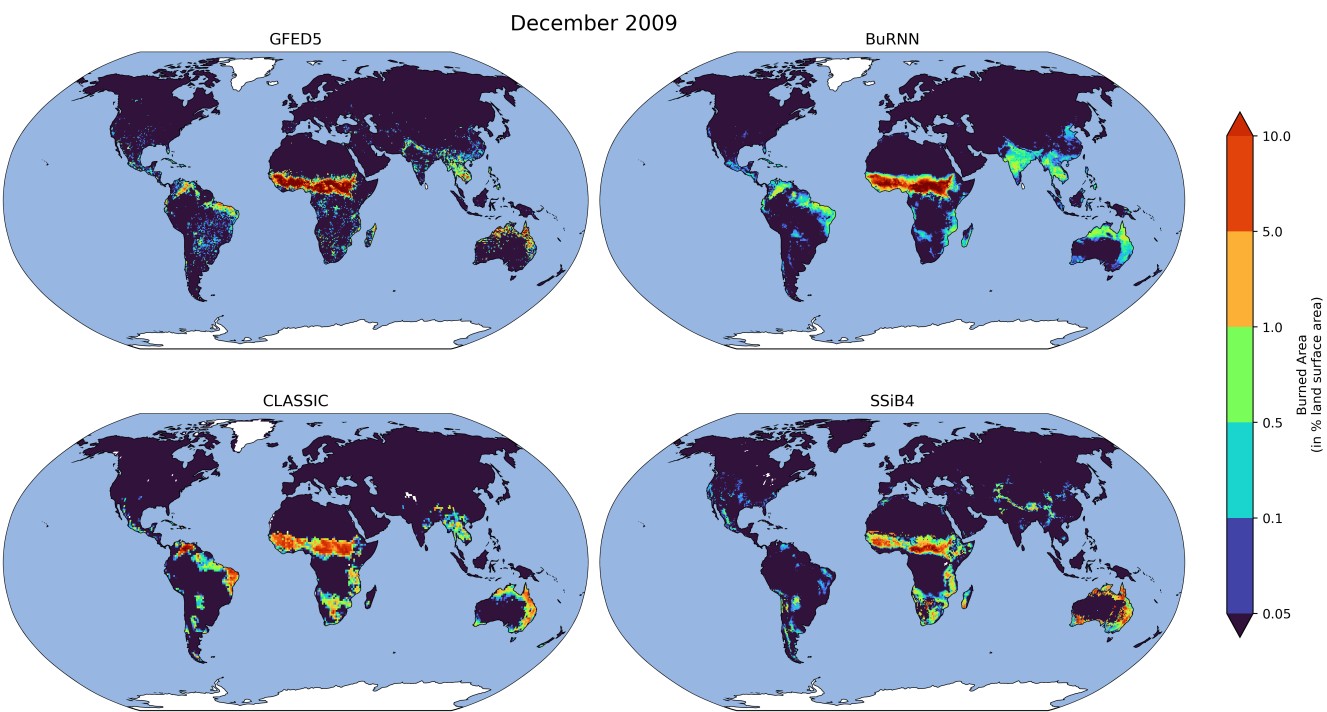

**Figure A6.** Comparison of BuRNN to GFED5 along with two process-based models (SSiB4 and CLASSIC) for December 2009.



**Table A1.** Models used for the calculation of the ISIMIP Biome characteristics.

|  | CLASSIC | ELM-ECA | DLEM | JULES-ES-VN6P3 | ORCHIDEE-MICT | SSiB4-TRIFFID-Fire | VISIT |
|---|---|---|---|---|---|---|---|
| cVeg |  | X |  |  |  | X | X |
| GPP | X | X | X | X | X | X | X |
| LAI | X | X |  |  | X | X | X |



**Table A2.** Evaluation scores of BuRNN and the FireMIP models in AUST. Colour scaling is based on the normalized values with the minimum and maximum values set to -2 and 2 (sigma). Better scores are marked in blue, while worse performance is in red.

| | BuRNN | CLASSIC | ELM-ECA | JULES-INFERNO-VN6P3 | LPJ-GUESS-SIMFIRE-BLAZE | LPJ-GUESS-SPITFIRE | LPJML5-7-10-FIRE | ORCHIDEE-MICT | SSIB4-TRIFFID-FIRE | VISIT |
|---|---|---|---|---|---|---|---|---|---|---|
| RMSE | 2.67 | 2.73 | 3.77 | 2.64 | 3.73 | 2.76 | 2.79 | 3.07 | 2.92 | 3.30 |
| Spatial RMSE | 0.96 | 0.94 | 1.14 | 0.67 | 0.79 | 0.73 | 0.98 | 1.18 | 0.96 | 1.48 |
| Correlation | 0.33 | 0.17 | 0.03 | 0.23 | 0.09 | 0.22 | 0.12 | 0.02 | 0.11 | -0.02 |
| Spatial Correlation | 0.57 | 0.37 | -0.04 | 0.79 | 0.56 | 0.63 | 0.33 | -0.01 | 0.28 | -0.03 |
| Monthly Correlation | 0.84 | 0.61 | 0.16 | 0.25 | 0.60 | 0.57 | 0.43 | 0.23 | 0.33 | 0.00 |
| Yearly Correlation | 0.91 | 0.34 | -0.21 | 0.27 | 0.23 | 0.57 | -0.16 | 0.76 | 0.64 | -0.30 |

**Table A3.** Evaluation scores of BuRNN and the FireMIP models in BOAS. Colour scaling is based on the normalized values with the minimum and maximum values set to -2 and 2 (sigma). Better scores are marked in blue, while worse performance is in red.

| | BuRNN | CLASSIC | ELM-ECA | JULES-INFERNO-VN6P3 | LPJ-GUESS-SIMFIRE-BLAZE | LPJ-GUESS-SPITFIRE | LPJML5-7-10-FIRE | ORCHIDEE-MICT | SSIB4-TRIFFID-FIRE | VISIT |
|---|---|---|---|---|---|---|---|---|---|---|
| RMSE | 1.20 | 1.29 | 2.28 | 1.28 | 1.30 | 1.32 | 1.27 | 1.31 | 1.26 | 1.48 |
| Spatial RMSE | 0.30 | 0.35 | 0.44 | 0.34 | 0.35 | 0.37 | 0.35 | 0.35 | 0.34 | 0.73 |
| Correlation | 0.43 | 0.05 | 0.07 | 0.09 | 0.07 | 0.04 | 0.15 | 0.03 | 0.20 | 0.01 |
| Spatial Correlation | 0.69 | 0.44 | 0.27 | 0.28 | 0.32 | -0.08 | 0.39 | 0.22 | 0.38 | 0.02 |
| Monthly Correlation | 0.90 | 0.09 | 0.50 | 0.17 | 0.50 | 0.04 | 0.69 | 0.03 | 0.72 | -0.16 |
| Yearly Correlation | 0.63 | 0.22 | 0.28 | -0.08 | -0.40 | -0.08 | 0.24 | 0.04 | -0.12 | 0.02 |





**Table A4.** Evaluation scores of BuRNN and the FireMIP models in BONA. Colour scaling is based on the normalized values with the minimum and maximum values set to -2 and 2 (sigma). Better scores are marked in blue, while worse performance is in red.

| | BuRNN | CLASSIC | ELM-ECA | JULES-INFERNO-VN6P3 | LPJ-GUESS-SIMFIRE-BLAZE | LPJ-GUESS-SPITFIRE | LPJML5-7-10-FIRE | ORCHIDEE-MICT | SSIB4-TRIFFID-FIRE | VISIT |
|---|---|---|---|---|---|---|---|---|---|---|
| RMSE | 0.41 | 0.43 | 1.30 | 0.41 | 0.50 | 0.50 | 0.42 | 0.57 | 0.42 | 0.90 |
| Spatial RMSE | 0.06 | 0.07 | 0.24 | 0.07 | 0.08 | 0.10 | 0.07 | 0.13 | 0.07 | 0.69 |
| Correlation | 0.18 | 0.05 | 0.05 | 0.06 | 0.02 | 0.04 | 0.03 | 0.00 | 0.09 | 0.02 |
| Spatial Correlation | 0.49 | 0.33 | 0.25 | 0.18 | 0.08 | 0.01 | 0.21 | -0.03 | 0.25 | 0.05 |
| Monthly Correlation | 0.55 | 0.46 | 0.45 | 0.63 | 0.58 | 0.62 | 0.40 | 0.39 | 0.67 | -0.03 |
| Yearly Correlation | 0.74 | 0.16 | 0.47 | 0.20 | -0.04 | 0.32 | 0.15 | -0.04 | 0.13 | -0.27 |

**Table A5.** Evaluation scores of BuRNN and the FireMIP models in CEAM. Colour scaling is based on the normalized values with the minimum and maximum values set to -2 and 2 (sigma). Better scores are marked in blue, while worse performance is in red.

| | BuRNN | CLASSIC | ELM-ECA | JULES-INFERNO-VN6P3 | LPJ-GUESS-SIMFIRE-BLAZE | LPJ-GUESS-SPITFIRE | LPJML5-7-10-FIRE | ORCHIDEE-MICT | SSIB4-TRIFFID-FIRE | VISIT |
|---|---|---|---|---|---|---|---|---|---|---|
| RMSE | 1.25 | 1.46 | 3.99 | 1.45 | 1.78 | 1.58 | 1.81 | 1.53 | 1.49 | 2.23 |
| Spatial RMSE | 0.37 | 0.57 | 1.23 | 0.67 | 0.56 | 0.64 | 0.65 | 0.61 | 0.62 | 1.60 |
| Correlation | 0.52 | 0.25 | 0.24 | 0.22 | 0.29 | 0.14 | 0.25 | -0.00 | 0.15 | -0.01 |
| Spatial Correlation | 0.63 | 0.22 | 0.18 | 0.23 | 0.31 | 0.02 | 0.14 | -0.11 | 0.04 | -0.16 |
| Monthly Correlation | 0.83 | 0.81 | 0.77 | 0.66 | 0.78 | 0.60 | 0.89 | 0.60 | 0.80 | 0.43 |
| Yearly Correlation | 0.79 | 0.66 | 0.67 | 0.20 | -0.16 | 0.13 | 0.74 | -0.12 | 0.54 | 0.23 |



**Table A6.** Evaluation scores of BuRNN and the FireMIP models in CEAS. Colour scaling is based on the normalized values with the minimum and maximum values set to -2 and 2 (sigma). Better scores are marked in blue, while worse performance is in red.

| | BuRNN | CLASSIC | ELM-ECA | JULES-INFERNO-VN6P3 | LPJ-GUESS-SIMFIRE-BLAZE | LPJ-GUESS-SPITFIRE | LPJML5-7-10-FIRE | ORCHIDEE-MICT | SSIB4-TRIFFID-FIRE | VISIT |
|---|---|---|---|---|---|---|---|---|---|---|
| RMSE | 1.16 | 1.25 | 1.94 | 1.23 | 1.35 | 1.33 | 1.49 | 1.65 | 1.28 | 1.46 |
| Spatial RMSE | 0.27 | 0.35 | 0.58 | 0.35 | 0.36 | 0.43 | 0.42 | 0.48 | 0.39 | 0.71 |
| Correlation | 0.35 | 0.08 | 0.02 | 0.08 | 0.04 | 0.00 | 0.02 | 0.09 | 0.09 | -0.00 |
| Spatial Correlation | 0.64 | 0.27 | -0.05 | 0.07 | 0.06 | -0.12 | -0.05 | 0.28 | 0.09 | -0.02 |
| Monthly Correlation | 0.76 | 0.39 | 0.25 | 0.34 | 0.56 | 0.20 | 0.40 | 0.25 | 0.50 | 0.00 |
| Yearly Correlation | 0.70 | 0.20 | 0.27 | -0.06 | 0.22 | 0.03 | -0.01 | 0.19 | 0.48 | 0.44 |

**Table A7.** Evaluation scores of BuRNN and the FireMIP models in EQAS. Colour scaling is based on the normalized values with the minimum and maximum values set to -2 and 2 (sigma). Better scores are marked in blue, while worse performance is in red.

| | BuRNN | CLASSIC | ELM-ECA | JULES-INFERNO-VN6P3 | LPJ-GUESS-SIMFIRE-BLAZE | LPJ-GUESS-SPITFIRE | LPJML5-7-10-FIRE | ORCHIDEE-MICT | SSIB4-TRIFFID-FIRE | VISIT |
|---|---|---|---|---|---|---|---|---|---|---|
| RMSE | 0.64 | 0.64 | 2.20 | 0.65 | 1.17 | 0.72 | 0.76 | 0.83 | 0.69 | 2.22 |
| Spatial RMSE | 0.20 | 0.20 | 0.66 | 0.22 | 0.32 | 0.24 | 0.21 | 0.30 | 0.17 | 2.11 |
| Correlation | 0.37 | 0.32 | 0.48 | 0.24 | 0.08 | 0.13 | 0.24 | 0.18 | 0.45 | -0.01 |
| Spatial Correlation | 0.47 | 0.21 | 0.62 | 0.03 | -0.02 | 0.07 | 0.17 | 0.22 | 0.61 | -0.06 |
| Monthly Correlation | 0.88 | 0.88 | 0.91 | 0.88 | 0.67 | 0.86 | 0.87 | 0.84 | 0.84 | 0.45 |
| Yearly Correlation | 0.89 | 0.94 | 0.89 | 0.93 | 0.51 | 0.91 | 0.94 | 0.93 | 0.94 | 0.72 |




**Table A8.** Evaluation scores of BuRNN and the FireMIP models in EURO. Colour scaling is based on the normalized values with the minimum and maximum values set to -2 and 2 (sigma). Better scores are marked in blue, while worse performance is in red.

| | BuRNN | CLASSIC | ELM-ECA | JULES-INFERNO-VN6P3 | LPJ-GUESS-SIMFIRE-BLAZE | LPJ-GUESS-SPITFIRE | LPJML5-7-10-FIRE | ORCHIDEE-MICT | SSIB4-TRIFFID-FIRE | VISIT |
|---|---|---|---|---|---|---|---|---|---|---|
| RMSE | 0.40 | 0.46 | 1.67 | 0.45 | 0.51 | 0.44 | 0.55 | 0.50 | 0.72 | 0.92 |
| Spatial RMSE | 0.11 | 0.15 | 0.48 | 0.17 | 0.15 | 0.15 | 0.18 | 0.17 | 0.36 | 0.75 |
| Correlation | 0.32 | 0.13 | 0.06 | 0.06 | 0.01 | 0.04 | 0.04 | 0.07 | 0.04 | 0.06 |
| Spatial Correlation | 0.63 | 0.24 | 0.11 | 0.03 | -0.01 | 0.08 | 0.06 | 0.15 | 0.05 | 0.14 |
| Monthly Correlation | 0.75 | 0.61 | 0.56 | 0.47 | 0.61 | 0.49 | 0.48 | 0.52 | 0.57 | 0.35 |
| Yearly Correlation | 0.50 | 0.14 | 0.20 | -0.18 | -0.33 | -0.07 | -0.13 | -0.03 | 0.26 | 0.27 |

**Table A9.** Evaluation scores of BuRNN and the FireMIP models in MIDE. Colour scaling is based on the normalized values with the minimum and maximum values set to -2 and 2 (sigma). Better scores are marked in blue, while worse performance is in red.

| | BuRNN | CLASSIC | ELM-ECA | JULES-INFERNO-VN6P3 | LPJ-GUESS-SIMFIRE-BLAZE | LPJ-GUESS-SPITFIRE | LPJML5-7-10-FIRE | ORCHIDEE-MICT | SSIB4-TRIFFID-FIRE | VISIT |
|---|---|---|---|---|---|---|---|---|---|---|
| RMSE | 0.30 | 0.31 | 0.52 | 0.34 | 0.62 | 0.74 | 1.04 | 0.69 | 0.56 | 0.71 |
| Spatial RMSE | 0.11 | 0.12 | 0.20 | 0.16 | 0.15 | 0.35 | 0.46 | 0.29 | 0.32 | 0.54 |
| Correlation | 0.19 | 0.02 | 0.03 | 0.05 | 0.00 | 0.00 | 0.05 | 0.06 | 0.07 | 0.01 |
| Spatial Correlation | 0.36 | 0.04 | 0.08 | 0.07 | -0.01 | -0.03 | 0.12 | 0.20 | 0.13 | 0.01 |
| Monthly Correlation | 0.79 | 0.63 | 0.61 | 0.67 | 0.63 | 0.46 | 0.57 | 0.61 | 0.72 | 0.24 |
| Yearly Correlation | 0.42 | -0.16 | 0.03 | -0.08 | 0.15 | 0.38 | -0.17 | 0.12 | -0.02 | -0.40 |





**Table A10.** Evaluation scores of BuRNN and the FireMIP models in NHAF. Colour scaling is based on the normalized values with the minimum and maximum values set to -2 and 2 (sigma). Better scores are marked in blue, while worse performance is in red.

| | BuRNN | CLASSIC | ELM-ECA | JULES-INFERNO-VN6P3 | LPJ-GUESS-SIMFIRE-BLAZE | LPJ-GUESS-SPITFIRE | LPJML5-7-10-FIRE | ORCHIDEE-MICT | SSIB4-TRIFFID-FIRE | VISIT |
|---|---|---|---|---|---|---|---|---|---|---|
| RMSE | 3.51 | 4.72 | 5.74 | 5.03 | 4.99 | 5.16 | 5.38 | 6.75 | 4.54 | 5.55 |
| Spatial RMSE | 1.01 | 1.50 | 2.25 | 1.96 | 2.09 | 2.27 | 2.03 | 3.37 | 1.79 | 2.64 |
| Correlation | 0.77 | 0.48 | 0.13 | 0.44 | 0.41 | 0.37 | 0.26 | 0.19 | 0.56 | 0.01 |
| Spatial Correlation | 0.93 | 0.76 | 0.44 | 0.65 | 0.69 | 0.44 | 0.59 | 0.31 | 0.69 | -0.06 |
| Monthly Correlation | 0.97 | 0.71 | 0.33 | 0.71 | 0.87 | 0.85 | 0.48 | 0.60 | 0.88 | 0.48 |
| Yearly Correlation | 0.74 | 0.93 | 0.38 | 0.88 | 0.62 | 0.58 | 0.44 | 0.00 | 0.76 | 0.32 |

**Table A11.** Evaluation scores of BuRNN and the FireMIP models in NHSA. Colour scaling is based on the normalized values with the minimum and maximum values set to -2 and 2 (sigma). Better scores are marked in blue, while worse performance is in red.

| | BuRNN | CLASSIC | ELM-ECA | JULES-INFERNO-VN6P3 | LPJ-GUESS-SIMFIRE-BLAZE | LPJ-GUESS-SPITFIRE | LPJML5-7-10-FIRE | ORCHIDEE-MICT | SSIB4-TRIFFID-FIRE | VISIT |
|---|---|---|---|---|---|---|---|---|---|---|
| RMSE | 0.68 | 2.86 | 2.52 | 0.82 | 1.39 | 1.61 | 1.29 | 1.13 | 0.86 | 1.04 |
| Spatial RMSE | 0.25 | 1.68 | 0.54 | 0.42 | 0.38 | 0.54 | 0.40 | 0.45 | 0.43 | 0.69 |
| Correlation | 0.63 | 0.56 | 0.35 | 0.46 | 0.29 | 0.30 | 0.32 | 0.35 | 0.45 | 0.02 |
| Spatial Correlation | 0.81 | 0.71 | 0.63 | 0.55 | 0.59 | 0.51 | 0.57 | 0.54 | 0.69 | -0.04 |
| Monthly Correlation | 0.93 | 0.93 | 0.83 | 0.80 | 0.85 | 0.83 | 0.84 | 0.75 | 0.84 | 0.67 |
| Yearly Correlation | 0.73 | 0.61 | 0.78 | 0.40 | 0.39 | 0.14 | 0.68 | -0.17 | 0.52 | 0.19 |





**Table A12.** Evaluation scores of BuRNN and the FireMIP models in SEAS. Colour scaling is based on the normalized values with the minimum and maximum values set to -2 and 2 (sigma). Better scores are marked in blue, while worse performance is in red.

| | BuRNN | CLASSIC | ELM-ECA | JULES-INFERNO-VN6P3 | LPJ-GUESS-SIMFIRE-BLAZE | LPJ-GUESS-SPITFIRE | LPJML5-7-10-FIRE | ORCHIDEE-MICT | SSIB4-TRIFFID-FIRE | VISIT |
|---|---|---|---|---|---|---|---|---|---|---|
| RMSE | 2.55 | 2.89 | 3.33 | 2.88 | 2.83 | 3.13 | 3.00 | 3.03 | 2.92 | 3.22 |
| Spatial RMSE | 0.88 | 1.06 | 1.07 | 1.02 | 1.01 | 1.23 | 1.06 | 1.04 | 1.11 | 1.40 |
| Correlation | 0.59 | 0.42 | 0.16 | 0.41 | 0.38 | 0.11 | 0.26 | 0.18 | 0.34 | 0.00 |
| Spatial Correlation | 0.64 | 0.55 | 0.41 | 0.54 | 0.55 | 0.19 | 0.41 | 0.46 | 0.41 | -0.08 |
| Monthly Correlation | 0.94 | 0.88 | 0.70 | 0.83 | 0.87 | 0.67 | 0.63 | 0.68 | 0.78 | 0.58 |
| Yearly Correlation | 0.54 | 0.30 | 0.68 | 0.34 | -0.09 | 0.13 | 0.54 | 0.16 | -0.02 | -0.31 |

**Table A13.** Evaluation scores of BuRNN and the FireMIP models in SHAF. Colour scaling is based on the normalized values with the minimum and maximum values set to -2 and 2 (sigma). Better scores are marked in blue, while worse performance is in red.

| | BuRNN | CLASSIC | ELM-ECA | JULES-INFERNO-VN6P3 | LPJ-GUESS-SIMFIRE-BLAZE | LPJ-GUESS-SPITFIRE | LPJML5-7-10-FIRE | ORCHIDEE-MICT | SSIB4-TRIFFID-FIRE | VISIT |
|---|---|---|---|---|---|---|---|---|---|---|
| RMSE | 3.43 | 4.61 | 5.85 | 4.75 | 4.56 | 4.86 | 4.89 | 6.87 | 4.02 | 5.39 |
| Spatial RMSE | 1.05 | 2.21 | 2.26 | 1.89 | 1.94 | 2.09 | 1.99 | 2.80 | 1.51 | 2.70 |
| Correlation | 0.74 | 0.45 | 0.15 | 0.42 | 0.46 | 0.33 | 0.45 | 0.09 | 0.61 | 0.02 |
| Spatial Correlation | 0.89 | 0.34 | 0.38 | 0.58 | 0.59 | 0.36 | 0.54 | 0.20 | 0.70 | -0.11 |
| Monthly Correlation | 0.99 | 0.95 | 0.44 | 0.77 | 0.96 | 0.88 | 0.91 | 0.52 | 0.97 | 0.49 |
| Yearly Correlation | 0.63 | 0.58 | 0.51 | 0.38 | -0.07 | 0.60 | 0.46 | 0.04 | 0.19 | 0.18 |



**Table A14.** Evaluation scores of BuRNN and the FireMIP models in SHSA. Colour scaling is based on the normalized values with the minimum and maximum values set to -2 and 2 (sigma). Better scores are marked in blue, while worse performance is in red.

| | BuRNN | CLASSIC | ELM-ECA | JULES-INFERNO-VN6P3 | LPJ-GUESS-SIMFIRE-BLAZE | LPJ-GUESS-SPITFIRE | LPJML5-7-10-FIRE | ORCHIDEE-MICT | SSIB4-TRIFFID-FIRE | VISIT |
|---|---|---|---|---|---|---|---|---|---|---|
| RMSE | 0.88 | 2.06 | 3.42 | 1.19 | 2.07 | 1.86 | 2.26 | 1.28 | 1.06 | 1.29 |
| Spatial RMSE | 0.29 | 1.22 | 0.83 | 0.68 | 0.66 | 0.92 | 0.96 | 0.54 | 0.44 | 0.70 |
| Correlation | 0.53 | 0.24 | 0.17 | 0.34 | 0.28 | 0.16 | 0.25 | 0.05 | 0.25 | 0.03 |
| Spatial Correlation | 0.67 | 0.25 | 0.35 | 0.47 | 0.44 | 0.21 | 0.31 | 0.06 | 0.21 | -0.04 |
| Monthly Correlation | 0.95 | 0.89 | 0.78 | 0.90 | 0.86 | 0.57 | 0.90 | 0.29 | 0.90 | 0.60 |
| Yearly Correlation | 0.74 | 0.71 | 0.51 | 0.48 | -0.12 | 0.67 | 0.53 | 0.55 | 0.51 | 0.16 |

**Table A15.** Evaluation scores of BuRNN and the FireMIP models in TENA. Colour scaling is based on the normalized values with the minimum and maximum values set to -2 and 2 (sigma). Better scores are marked in blue, while worse performance is in red.

| | BuRNN | CLASSIC | ELM-ECA | JULES-INFERNO-VN6P3 | LPJ-GUESS-SIMFIRE-BLAZE | LPJ-GUESS-SPITFIRE | LPJML5-7-10-FIRE | ORCHIDEE-MICT | SSIB4-TRIFFID-FIRE | VISIT |
|---|---|---|---|---|---|---|---|---|---|---|
| RMSE | 0.38 | 0.78 | 2.26 | 0.65 | 1.04 | 0.55 | 0.91 | 0.52 | 0.55 | 0.95 |
| Spatial RMSE | 0.10 | 0.31 | 0.66 | 0.36 | 0.32 | 0.23 | 0.33 | 0.18 | 0.23 | 0.74 |
| Correlation | 0.15 | 0.04 | 0.04 | 0.05 | 0.06 | 0.01 | 0.03 | -0.01 | 0.06 | -0.00 |
| Spatial Correlation | 0.38 | 0.18 | 0.09 | 0.16 | -0.09 | -0.09 | -0.02 | -0.04 | 0.04 | -0.05 |
| Monthly Correlation | 0.73 | 0.28 | 0.19 | 0.14 | 0.22 | 0.14 | 0.09 | 0.10 | 0.29 | -0.04 |
| Yearly Correlation | 0.66 | 0.64 | 0.47 | 0.82 | -0.15 | 0.53 | 0.44 | 0.72 | 0.73 | -0.08 |





**Table A16.** Regional correlation of annual burned area of BuRNN and FireCCiLT11 (1982-1993 and 1997-2018) and between FireCCiLT11 and GFED5 (1997-2018).

|  | BuRNN-FireCCiLT11: 1982-1993 | BuRNN-FireCCiLT11: 1997-2018 | GFED5-FireCCiLT11: 1997-2018 |
|---|---|---|---|
| BONA | 0.62 | -0.13 | -0.07 |
| EURO | 0.58 | 0.21 | 0.05 |
| MIDE | -0.26 | -0.44 | 0.29 |
| BOAS | 0.20 | -0.20 | -0.04 |
| TENA | 0.27 | 0.22 | -0.23 |
| CEAS | -0.28 | 0.30 | 0.29 |
| CEAM | 0.60 | 0.02 | 0.12 |
| SEAS | 0.04 | -0.21 | 0.32 |
| NHSA | 0.06 | 0.08 | 0.24 |
| EQAS | -0.15 | -0.07 | -0.06 |
| SHSA | 0.21 | -0.05 | 0.23 |
| NHAF | 0.21 | 0.36 | 0.52 |
| SHAF | 0.52 | 0.50 | -0.06 |
| AUST | 0.32 | 0.22 | 0.19 |





**Table A17.** Regional evaluation scores of BuRNN. Colour scaling has been done based on the ranked values compared to the with the minimum RMSE and maximum correlations coloured blue and the highest RMSE and correlation coloured red.

| | AUST | BOAS | BONA | CEAM | CEAS | EQAS | EURO | MIDE | NHAF | NHSA | SEAS | SHAF | SHSA | TENA |
|---|---|---|---|---|---|---|---|---|---|---|---|---|---|---|
| RMSE | 2.78 | 0.95 | 0.42 | 0.53 | 1.10 | 0.46 | 0.29 | 0.23 | 2.97 | 0.76 | 1.49 | 3.32 | 0.93 | 0.45 |
| Spatial RMSE | 0.84 | 0.18 | 0.05 | 0.24 | 0.22 | 0.23 | 0.08 | 0.08 | 0.86 | 0.27 | 0.55 | 0.91 | 0.27 | 0.09 |
| Correlation | 0.28 | 0.34 | 0.08 | 0.33 | 0.21 | 0.31 | 0.30 | 0.16 | 0.71 | 0.54 | 0.35 | 0.67 | 0.42 | 0.08 |
| Spatial Correlation | 0.55 | 0.61 | 0.20 | 0.40 | 0.43 | 0.40 | 0.59 | 0.29 | 0.86 | 0.70 | 0.48 | 0.84 | 0.58 | 0.18 |
| Monthly Correlation | 0.81 | 0.86 | 0.25 | 0.84 | 0.47 | 0.91 | 0.76 | 0.78 | 0.95 | 0.82 | 0.80 | 0.97 | 0.89 | 0.60 |
| Yearly Correlation | 0.92 | 0.73 | 0.51 | 0.70 | 0.75 | 0.96 | 0.52 | 0.46 | 0.66 | 0.62 | 0.28 | 0.27 | 0.94 | 0.67 |

best                                          average                                          worst



*Code and data availability.* All code for the pre-processing, training and post-processing of BuRNN is openly accessible on GitHub (https://github.com/VUB-HYDR/BuRNN)) and is archived on Zenodo under copyright license CC BY 4.0 (https://doi.org/10.5281/zenodo.16419609;

Lampe, 2025a). The 1901-2019 burned area simulation of BuRNN is available on Zenodo along with all pre-processed data to train BuRNN (https://zenodo.org/records/16918071; Lampe, 2025b). GFED5, HistLight and WGLC can be retrieved originally from Zenodo as well (https://zenodo.org/records/7668424, https://zenodo.org/records/6405396 and https://zenodo.org/records/15215319; Chen et al., 2023a; Kaplan and Lau, 2022b; Kaplan, 2025). The original ISIMIP data is available through the ISIMIP data repository (https://data.isimip.org/), the authentic SPEI data from SPEIbase can be downloaded from https://spei.csic.es/database.html. The CLM data is automatically generated

during the pre-processing for a CLM model run.

*Author contributions.* SL, WT and EC conceptualized the study. SL, LG, BK, VH and BLS designed the model architecture. SL programmed and trained the model. SL, WT, LG, BK, SH and DK performed the analysis. WT supervised the project. All authors contributed to the final version of the manuscript.

*Competing interests.* The authors declare that they have no conflict of interest.

*Acknowledgements.* S.L. was supported by a PhD Fundamental Research Grant by Fonds Wetenschappelijk Onderzoek-Vlaanderen (FWO; 11M7725N). The computational resources and services used in this work were provided by the VSC (Flemish Supercomputer Center), funded by the FWO and the Flemish Government. W.T. acknowledges funding from the European Research Council (ERC) under the European Union's Horizon Framework research and innovation programme (grant agreement No 101076909; ERC Consolidator Grant 'LACRIMA'). ECh has been funded by the European Space Agency's Climate Change Initiative (ESA CCI) programme (Contract No. 4000126706/19/I-

NB). S.H. acknowledges support from the Max Planck Tandem group programme and from Universidad del Rosario within the programme of Fondos de arranque.



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
