# Peer review of "BuRNN (v1.0): A Data-Driven Fire Model"

_EGUsphere, 2025_

## Author Response (AR1)

**Reviewer 1**

The manuscript presents a monthly 0.5° machine-learning emulator of burned area trained on GFED5 with region-blocked cross-validation. It uses a Long Short-Term Memory (LSTM) architecture to capture temporal dependence on the input features. The trained BuRNN model is then applied to reconstruct global burned area for 1901–2019, and the simulations are compared against independent long-term fire observation databases from five countries and the EU. Overall, the manuscript is well written, and the trained model shows promising results. Yet several core claims are insufficiently supported. In particular, BuRNN is target-aligned to GFED5 when compared with process models, which makes the comparison unfair. Model biases and observational uncertainties must be clearly distinguished. The transferability of a model trained on recent conditions to historical periods requires further discussion. The interpretability of the SHAP analysis assumes feature independence, which is strongly violated; this needs further evaluation.

> We thank the reviewer for their time and effort in reviewing our work. Partly thanks to their suggestions, we have managed to improve the quality of BuRNN substantially. We have made several impactful changes, which we summarise here below, before responding to each of the reviewer's comments in detail.

> - We now also normalize the targets (after applying the log-transform)
> - We have removed several features (SPEI and the subdivision into different regional C4 grasses)
> - We have altered the loss function from MSE to a combination of Binary Cross Entropy (classifier for fire occurrence) and Gaussian Negative Log Likelihood (Gaussian NLL). We now output three variables, the first is used in the classifier, while the remaining two are used in the Gaussian NLL.
> - We have added a comparison between GFED5 and FireCCI51 products
> - We have changed from SHAP to Integrated Gradients

> All this has led to significant improvements in model performance, BuRNN now accurately captures mean burned area, interannual variability and seasonality in almost all places.

**Major comments**

1. Because BuRNN is trained on GFED5 while process-based models are not, any superiority shown for 1997–2019 primarily reflects alignment to the chosen satellite target rather than absolute accuracy. The manuscript notes this when introducing the FireCCI51 cross-check ("we repeat this evaluation… using the 2001–2019 FireCCI51 dataset… because our model is specifically trained to predict GFED5…"), yet the framing still concludes that BuRNN outperforms process models (see Table A17). Where BuRNN appears closer to FireCCI51 than the process models, this likely indicates that GFED5 and FireCCI51 are more mutually consistent than FireCCI51 is with the process models, not that BuRNN is more accurate in an absolute sense.

> We thank the reviewer for this detailed comment. Indeed, training our model to predict GFED5 burned area and then evaluating its performance against it favors BuRNN. However, GFED5 is now

recognized as the most reliable dataset available. Moreover, even though the process-based models aren't directly trained to predict GFED5-type burned area, they are (almost) all parametrized to predict GFED4/GFED4.1s-type burned area. In each of the model evaluation papers, the process-based fire models are compared to GFED4/GFED4.1s (Mangeon et al. (2016), Huang et al. (2020), Oberhagemann et al. (2025)). By comparing both BuRNN and the process-based models to FireCCI5.1, we have a 'neutral' common ground to compare performance against. With the improvements added to this version, BuRNN now lies more closely to GFED5 than FireCCI5.1. Nonetheless, it still mostly outperforms the process-based models in many correlation metrics compared to FireCCI5.1. However, we understand the need for taking the observational uncertainty into account. Therefore, we have now added comparisons between the observational products in Supplementary Table A2 and discuss these in the Results section (Section 3.1). Inter-observational performance informs about maximum achievable performance given observational uncertainty between products i.e., at what point does 'improving' with respect to single data source not make sense anymore and is the model indiscernible from observational products. In this regards, BuRNN is now globally within observational uncertainty for spatial correlation, monthly correlation and spatial RMSE.

2. Comparing Figure 4 (regional/global annual BA time series) with Figure A2 (BuRNN vs FireCCiLT11), BuRNN aligns more closely with FireCCiLT11 than with GFED5 during the overlap period, indicating that a meaningful portion of the discrepancy reflects observational product differences rather than model bias. The same issue arises in national/EU validations. Because EFFIS explicitly excludes cropland fires, BuRNN–EFFIS agreement or disagreement is interpretable only after restricting BuRNN/GFED5 to the non-cropland domain; otherwise systematic differences over cropland regions are expected by construction.

The updated version of BuRNN now lies much closer to GFED5 than to FireCCiLT11. As mentioned in response to Comment 1, we now also include inter-observational uncertainty to our evaluation metrics. However, we believe including the EFFIS burned area still makes sense for many countries as cropland fires represent 'only' ~20% of total area burned in Europe, implying that a systematic error due to cropland fires will be of this order at most (San-Miguel-Ayanz et al. (2023), (2024), (2025)). Countries with notable (up to one third of total burned area) crop fires are typically Romania, Bulgaria and Italy.

3. Methods state that SHAP values are computed "relative to the average prediction over a dataset," using GradientExplainer with integration from the average input to each sample. The manuscript does not specify whether "the dataset" and the average input are defined globally or per region, yet regional SHAP panels are compared across regions. If a global background is used, a regional bar reflects deviation from a global reference rather than the region's climatology; if the background is recomputed per region, cross-region bar magnitudes are not directly comparable.

We have now changed our XAI method from SHAP to Integrated Gradients (IG). However, IG also requires a baseline, which we have chosen as the global mean. This makes regions inter-comparable to each other. We also considered using the regional means, but this prevents any statements on differences between regions in terms of feature importance. We now more clearly state what the baseline is exactly in Section 2.4.

"IG compares the prediction at an input x to the prediction with a reference baseline input $x_0$ and integrates the model's gradients along a straight-line path between them. Here, we applied the global mean for each feature as baseline. So the IG results need to be interpreted as 'How strong does each feature affect burned area in this region compared to the global mean of this feature'. One caveat of this approach is that if a feature in a particular region tends to be close to the global mean, then attribution for that feature will be low as the integration between sample and baseline will be performed over a short path."

4. FWI, tasmax, SPEI, ET/GPP and related variables are strongly dependent by construction. Under such collinearity, SHAP can redistribute attribution among correlated predictors, so statements like "FWI is most important everywhere" are not stable. Although the manuscript cautions that SHAP is not causal and is difficult to interpret with interdependent inputs, the subsequent discussion reads mechanistically from the bars. This is inconsistent with the stated caveat.

We have now removed SPEI (EvapoTranspiration was never included), we have also changed our XAI method from SHAP to Integrated Gradients (IG). However, the reviewers main point still stands, IG also redistributes among correlated features. We now explicitly mention this in the text in Section 2.4.

"Lastly, highly correlated features will have their attributed importance spread across each other and hence be lower than if only a single of these features was provided."

5. The reconstruction extends BuRNN to 1901–2019 and is compared with FireCCiLT11 for 1982–2018. Applying a mapping trained on 1997–2019 to 1901–1996 introduces non-stationarity risk in regions with large changes in ignition, suppression or land-use practices. For instance, should we expect human play a same current role in the 1910s? In addition, vegetation predictors (GPP, LAI, cVeg) are ISIMIP DGVM ensemble means, which predicted with a process-based fire model. The manuscript should discuss how biases in the vegetation inputs and the omission of fire-vegetation feedbacks in BuRNN affect its predictions.

We agree with the reviewer on these points. However, this is something we cannot solve with the available data at hand, process-based models suffer from the same issues. We have added these points in the discussion:

"However, three main sources of uncertainty and drawbacks need to be raised. First, our model will learn relationships between population densities, GDP and fire occurrence. These might have changed over the last 120 years and nor BuRNN, nor the process-based models can account for this currently. Secondly, BuRNN also relies on three inputs from DGVMs, which are of course reliant on the performance of the model ensembles for these variables. Lastly, in BuRNN there are currently no fire-vegetation feedbacks, which are present in most process-based models.

**Minor comments**

- In the introducing. While the satellite products suffer uncertainties due to cloud cover, spatial resolution, rapid regrowth and obscuration by unburned vegetation, etc, the process-based model or machine model calibrated or trained on satellite-based product inherits those uncertainties.

That is correct, we have now added this to the introduction.

> "Moreover, satellite observations suffer uncertainties due to cloud cover, spatial resolution, rapid regrowth and obscuration by unburned vegetation, which are propagated further into modelling efforts."

- Since many datasets across different period are compared, the figure captions can be enhanced with better notion of time period, dataset, and units. For instance, confusion can be arised when looking at Fig. 3 which shows the burned area fraction (%) vs Fig. 4 showing burned area in Mha

We have updated the captions for Figures 2, 3, 4, 6 and 7 to include time period, dataset(s) and units.

- Line 132, Burned area is right-skewed and zero-inflated. How are zero burned area handled under the log transform? Was any zero-handling or sampling strategy, or a distribution-/tail-aware loss is applied. A plot of the empirical distribution (PDF/CDF) by region would help evaluate if any especial treatment is needed.

We apply numpy's log1p transform i.e., log(1 + x). We now also apply a new loss, which predicts a distribution (mean and std) of burned area rather than a single value. We don't implement any special zero-aware sampling strategy. We have added a histplot showing the observed and modelled burned area (in % land surface area) distributions for each region. Note that the y-scale is log-scaled. We have added this to the manuscript as Supplementary Figure A23.

[Figure]

- Line 261, Line 261, Underestimation of interannual variability in several regions and suggests VPD/ET/PET as potential drivers. It would be worth to include them to retrain the model.

  We thank the reviewer for this suggestion, although we have not yet included VPD/ET/PET as potential drivers, we will likely try this out for the next version of BuRNN. However, the underestimation of interannual variability has now largely been resolved.

**References**

Mangeon, S., Voulgarakis, A., Gilham, R., Harper, A., Sitch, S., & Folberth, G. (2016). INFERNO: A fire and emissions scheme for the UK Met Office's Unified Model. *Geoscientific Model Development*, *9*(8), 2685-2700.

Huang, H., Xue, Y., Li, F., & Liu, Y. (2020). Modeling long-term fire impact on ecosystem characteristics and surface energy using a process-based vegetation-fire model SSiB4/TRIFFID-Fire v1. 0. *Geoscientific Model Development Discussions*, *2020*, 1-41.

Oberhagemann, L., Billing, M., von Bloh, W., Drüke, M., Forrest, M., Bowring, S. P., ... & Thonicke, K. (2025). Sources of uncertainty in the SPITFIRE global fire model: development of LPJmL-SPITFIRE1. 9 and directions for future improvements. *Geoscientific Model Development*, *18*(6), 2021-2050.

San-Miguel-Ayanz, J., Durrant, T., Boca, R., Maianti, P., Libertà, G., JACOME, F. O., ... & LOFFLER, P. (2023). *Forest Fires in Europe, Middle East and North Africa 2022*.

San-Miguel-Ayanz, J., Durrant, T., Boca, R., Maianti, P., Libertá, G., Oom, D., ... & Broglia, M. (2024). *Advance report on forest fires in Europe, Middle East and North Africa 2023*.

San-Miguel-Ayanz, J., Durrant, T., Boca, R., Maianti, P., Libertá, G., Oom, D., ... & Sedano, F. (2025). *Advance report on forest fires in Europe, Middle East and North Africa 2024*.

**Reviewer 2**

Lampe et al. developed a data-driven fire model (BuRNN) based on Long Short-Term Memory to estimate global gridded burned area from 1901 to 2019. BuRNN was trained with satellite-based burned area (GFED5) multiple climates, land cover, vegetation dynamics, and socioeconomic datasets during 2001 to 2020. The trained BuRNN was then used to reconstruct the burned area from 1901 to 2019. Indeed, wildfire remains to be highly uncertain in process-based models. Data-driven methods have the potential to improve the estimation of wildfire, which has been shown in previous studies. This study contributes to our understanding of wildfire mechanisms by providing long-term global burned area estimation and factor importance analysis. However, I have some concerns about the reliability of the data-driven model. Please find the details of the major and specific comments in the following. I recommend a major revision for the current manuscript version.

We thank the reviewer for their time and effort in reviewing our work. Partly thanks to their suggestions, we have managed to improve the quality of BuRNN substantially. We have made several impactful changes, which we list here below, before responding to each of the reviewer's comments.

- We now also normalize the targets (after applying the log-transform)
- We have removed several features (SPEI and the subdivision into different regional C4 grasses)
- We have altered the loss function from MSE to a combination of Binary Cross Entropy (classifier for fire occurrence) and Gaussian Negative Log Likelihood (Gaussian NLL). We now output three variables, the first is used in the classifier, while the remaining two are used in the Gaussian NLL.
- We have changed from SHAP to Integrated Gradients

All this has led to improvements in model performance, BuRNN now accurately captures mean burned area, interannual variability and seasonality in almost all places.

**Major Comments**

1. BuRRN consistently underestimate the burned area across different regions and at global scales. This underestimation seems to be systematic bias, however, I don't understand why such error cannot be reduced in BuRRN, which is built on LSTM. I suppose machine learn approaches (including LSTM) is effective in reducing systematic bias. Please take a detailed analysis on this error. I suggest showing the validation at monthly scales. One possible reason is that BuRRN underestimate the maximum monthly burned area, which it may capture lower values well.

BuRNN was indeed underestimating burned area, something which was bothering us as well. We have been able to now overcome this by normalizing the target data based on training + validation set mean and standard deviation. This causes a minor data leakage between training and validation

set, but ensures consistency between folds for the same test set and still prevents the models from seeing any data about the test set during training.

We also followed the reviewer's suggestion and added in an extra figure at monthly scales. We have added in Figure 7, showing a monthly comparison of BuRNN and GFED5 for 2010-2015 and Supplementary Figure A23, showing histograms of regionally observed and predicted burned area (in % land surface area). Together with information that was already provided e.g., evaluation tables listing monthly correlation and Supplementary Figures A4-A6, we hope this provides a clear understanding of BuRNN's capabilities and limitations.

2.  There is a lack of validation of temporal extrapolation. The authors did the out-of-sample validation by splitting the global domain to 11 subregions. However, they didn't split the dataset in time to validate the model performance in temporal extrapolation. As shown in Figure 4, BuRNN cannot capture the annual trend presented in the benchmark dataset (i.e., GFED5). One of the objectives of this study is to reconstruct global burned area from 1901 to 2019. The failure of capturing the sensitivity to the environment changes during training period may suggest the reconstruction in the past is highly uncertain.

We thank the reviewer for this important remark. The updated version of BuRNN is able to capture the annual trend in most regions (and globally) relatively well now. We have calculated Theil-Sen sloped for the 2003-2019 burned areas. BuRNN now estimates an annual decrease of -4.78 ± 2.21 (mean ± 2SD) Mha per year globally compared to -8.45 ± 3.09 Mha per year from GFED5. We have added these results and the results per region to a new Supplementary Table A2.

We have also followed the reviewer's suggestion and trained BuRNN with the regional folds on the 1997-2015 data and evaluated the 2016-2019 period. BuRNN performed quasi-exactly the same as it does when it has seen the entire time period (see Figure below). The only difference was a small (~50 Mha/per year) increase in mean burned area globally. This is explained by the trend in the GFED5 observed burned area. There is a global negative trend in burned area, so the early period has a slightly higher mean burned area than the last few years. This issue would resolve itself if we do a one or two year temporal fold instead of four years. We considered turning the regional folds into temporal and regional folds. However, this would increase the cost of training and running BuRNN by K-fold (with K = the number of temporal folds, so 10-20). As speed is one of the advantages of BuRNN we opted to not do this. We do sincerely thank the reviewer for this suggestion as it was a valid and important question to ask.

[Figure]

*Regional timeseries when BuRNN is trained on the 1997-2015 period.*

**Specific Comments**

- Line 29: Please specify how the burn area (i.e. 3.5 – 4.5 million km^2) was estimated. As in later sentence, the authors provide the burn area from satellite observation, it is helpful to know how the previous estimation was made.

This is also based on satellite observations from MODIS i.e., the GFED4.1s and FireCCI5.1 products (the two citations at the end of the sentence in the manuscript). However, we know that these underestimate the actual burned area because their spatial resolution misses many small fires (e.g., Chuvieco et al. (2022)), GFED5 tries to account for this and likely gives a more realistic total area burned.

- Figure A1: Please specify which color represent training.

We have now expanded the caption beneath figure A1. It now reads as follows:

"Division of the 43 regions into 11 folds, used for training the models. The regions marked in yellow represent the 3-4 AR6 regions in that fold. During training we set each fold aside once, then train

5 models on the remaining 10 folds, each time with 8 folds as training and 2 folds as validation. E.g., Fold 1 is set aside as testing fold, then folds 2-3 are used as validation and folds 4-11 as training. Then, folds 4-5 are used as validation and folds 2-3 and 6-11 as training. This is followed by folds 6-7 as validation and folds 2-5 and 8-11 as training, etc."

- Line 118: Why not randomly grouping the regions into 11 folds?

Randomly grouping the regions can result in cases where two adjacent regions are in the same fold. This is not a problem per se if there are multiple similar ecoregions in the world. However, in some cases e.g., Eastern Canada and Western Canada, you'd like to ensure that both regions are not in the same fold so that dynamics can be learned from one region and applied to the other. By manually dividing the 43 regions into 11 diverse folds we ensure that each fold contains a diverse mix of fire regions, which maximizes the learning potential of our models.

- Line 141: Is 3 years the sequence length used in LSTM? Please elaborate how the sequence length is determined.

During each epoch, each spatial sample (pixel) is passed once. However, instead of passing the entire timeseries, we pass a random slice of 6 years to the LSTM and only score it for the predictions made on the last 3 years. In the next epoch, for the same sample, another random slice of 6 years will be passed along. We pass 6 years so the model gets 3 years to get its internal memory stabilised/up-to-date with the real-world conditions. The spinup length of 3 years was chosen based on real-world fire process understanding, while the 3 year sequence length was selected as a decent choice for model convergence. A shorter sequence length makes epochs faster, but leads to more epochs being needed to train the model until convergence and vice versa. We have now clarified this in the text:

> "During training we ignore the first 36 predictions (3 years) to allow the LSTM's memory state to spin up and then evaluate the predictions of the following 3 years using a custom loss function. In each epoch, we pass each pixel/location in the training set once and randomly select a 6 year time slice. The spinup period of 3 years was chosen based on fire-process understanding and the prediction length of 3 years was chosen in function of model convergence speed."

- Line 155: Could the authors provide some explanation on how the SHAP handle the correlation between input variables?

We have changed our XAI approach from SHAP to IntegratedGradients. In short, the approach does not account for correlated features and spreads feature importance across them. We now explicitly mention this in the text in Section 2.4.

> "Lastly, highly correlated features will have their attributed importance spread across each other and hence be lower than if only a single of these features was provided."

- Line 214: Are those process-based models calibrated? If not, it is not surprise that data-driven method outperforms the process-based model.

Yes and no, most of them are not calibrated or bias-corrected directly, but many/all of them have been parameterised so they match the previous version of GFED (GFED4/GFED4.1s) as close as possible. For many years GFED4/GFED4.1s was considered the state of the art and was consequently used to validate the fire models performance e.g., Mangeon et al. (2016), Huang et al. (2020), Oberhagemann et al. (2025). The GFED4.1s and FireCCI5.1 (the secondary dataset we use for evaluating BuRNN's performance against the process-based models performance) products have a similar total annual burned area so should favour most of the process-based models.

- Line 219: "between the FireMIP models" is repeated.

We have removed the duplicate, thank you for spotting this mistake.

- Line 223: The evaluation for the 14 fire regions is not consistent with the global monthly evaluation. According to Figure 3, the BuRNN bias does not seems to be significantly. However, Figure 4 shows BURNN significantly underestimated annual burned area at both regional and global scales. Please explain this inconsistency. In addition, BURNN systematically underestimate the burned area, which I don't understand why the LSTM cannot resume this error.

BuRNN no longer has the consistent underestimation so we believe most of this comment has been addressed. As to the observed discrepancy between Figures 3 and 4; both plots are correct. There was indeed an underestimation as Figure 3 showed, in Figure 4 one can see that the distribution plot in the upper-left corner is left-tailed indicating more underestimation than overestimation. Moreover, several regions (mainly in Africa) have pixels where burned area is 2-3% of the land surface lower than observed, which is the cause of the total underestimation. We believe the issue might have been more an 'optical effect' because the process-based models generally have larger under/over estimations of burned area, making the under/over estimations of BuRNN seem small.

- Figure 4: I suggest the authors to report the slope for the annual burn area for both GFED5 and BuRNN to validate if BuRNN's can simulate the annual trend. This will demonstrate if BuRRN can capture the sensitivity of wild fire to the changing climate in the past. For example, the global plot in Figure 4 shows GFED5 exhibit a decreasing trend, while BuRNN exhibit no trend.

We now report the observed (-8.45 ± 3.09) and modelled (-4.78 ± 2.21) Theil-Sen slope for 2003-2019 in the figure caption. Additionally, we also provide the slopes for each of the GFED regions in the new Supplementary Table A2.

- Line 227: I cannot agree with "excellently modelled". Those regions are better than other regions, but they are still underestimated in BuRRN.

This has now likely been resolved through the improved performance of BuRNN.

- Line 228: "BuRNN captures the pattern of the interannual variability well, but consistently underestimates the amplitude and total burned area," This statement is repetitive, which is reported earlier in this paragraph.

We have removed the statement.

- Line 245: "in MIDE, BuRNN…"

We have adjusted this, thank you.

- Line 248: Could the authors try to explain why BuRNN has this behavior: "e higher the average monthly burned area in a region, the better/easier predictions are for that region"?

It is less of a problem now, but in regions that see little burning it can be hard for the model to figure out why it is only burning at that specific time/location. We have removed the statement.

- Line 262: Why not including those variables in BuRNN if they are considered as important? There are existing global datasets of evapotranspiration can be used.

For the next version of BuRNN we will likely try out adding some of these variables, but for now the issues stated above that sentence have been resolved. Consequently, we have removed the statement.

- Figure 5: The current text is not very clear to see, please consider increasing the font size. Is the feature importance plotted as Volin plot? I suggest changing to box plot, which gives the 25th, 50th, and 75th percentile. The current plot is dominated by the outliers, which does not help to interpret the data.

We have altered these plots as we now use Integrated Gradients instead of SHAP.

- Line 269: Why C4 grass Africa is considered as a feature for EURO, BOAS, SHSA, and GLOBAL? The absence of African C4 grass for explaining the lower burned area does not make sense to me. Shouldn't we focus on existing land covers in each region for the feature importance analysis?

We have now removed the division into American, African and Eurasia + Oceania C4 grasses.

- Line 295: Please report the annual correlation.

We have added the annual correlation for 1997-2017 in brackets to the text now (0.29).

- Line 298 – Line 302: If EFFIS does not include cropland fire, I don't see a point to benchmark BuRRN with it.

In most European countries cropland fires don't exist anymore or are very minimal. We believe including the EFFIS burned area still makes sense for many countries as cropland fires represent 'only' ~20% of total area burned in Europe. Countries with notable (up to one third of total burned area) crop fires are typically Romania, Bulgaria and Italy.

References

Chuvieco, E., Roteta, E., Sali, M., Stroppiana, D., Boettcher, M., Kirches, G., ... & Albergel, C. (2022). Building a small fire database for Sub-Saharan Africa from Sentinel-2 high-resolution images. *Science of the total environment*, *845*, 157139.

Mangeon, S., Voulgarakis, A., Gilham, R., Harper, A., Sitch, S., & Folberth, G. (2016). INFERNO: A fire and emissions scheme for the UK Met Office's Unified Model. *Geoscientific Model Development*, *9*(8), 2685-2700.

Huang, H., Xue, Y., Li, F., & Liu, Y. (2020). Modeling long-term fire impact on ecosystem characteristics and surface energy using a process-based vegetation-fire model SSiB4/TRIFFID-Fire v1. 0. *Geoscientific Model Development Discussions*, *2020*, 1-41.

Oberhagemann, L., Billing, M., von Bloh, W., Drüke, M., Forrest, M., Bowring, S. P., ... & Thonicke, K. (2025). Sources of uncertainty in the SPITFIRE global fire model: development of LPJmL-SPITFIRE1. 9 and directions for future improvements. *Geoscientific Model Development*, *18*(6), 2021-2050.

San-Miguel-Ayanz, J., Durrant, T., Boca, R., Maianti, P., Libertà, G., JACOME, F. O., ... & LOFFLER, P. (2023). *Forest Fires in Europe, Middle East and North Africa 2022*.

San-Miguel-Ayanz, J., Durrant, T., Boca, R., Maianti, P., Libertá, G., Oom, D., ... & Broglia, M. (2024). *Advance report on forest fires in Europe, Middle East and North Africa 2023*.

San-Miguel-Ayanz, J., Durrant, T., Boca, R., Maianti, P., Libertá, G., Oom, D., ... & Sedano, F. (2025). *Advance report on forest fires in Europe, Middle East and North Africa 2024*.